# LookaheadKV: Fast and Accurate KV Cache Eviction by Glimpsing into the Future without Generation

**Jinwoo Ahn**[*]  **Ingyu Seong**[*]  **Akhil Kedia**  **Junhan Kim**  **Hyemi Jang**
**Kangwook Lee**[†]  **Yongkweon Jeon**[†]

Samsung Research
{jinwoo.ahn, ingyu.seong, kw.brian.lee, dragwon.jeon}@samsung.com

## Abstract

Transformer-based large language models (LLMs) rely on key–value (KV) caching to avoid redundant computation during autoregressive inference. While this mechanism greatly improves efficiency, the cache size grows linearly with the input sequence length, quickly becoming a bottleneck for long-context tasks. Existing solutions mitigate this problem by evicting prompt KV that are deemed unimportant, guided by estimated importance scores. Notably, a recent line of work proposes to improve eviction quality by "glimpsing into the future", in which a draft generator produces a surrogate future response approximating the target model's true response, and this surrogate is subsequently used to estimate the importance of cached KV more accurately. However, these approaches rely on computationally expensive draft generation, which introduces substantial pre-filling overhead and limits their practicality in real-world deployment. To address this challenge, we propose LookaheadKV, a lightweight eviction framework that leverages the strength of surrogate future response without requiring explicit draft generation. LookaheadKV augments transformer layers with parameter-efficient modules trained to predict true importance scores with high accuracy. Our design ensures negligible runtime overhead comparable to existing inexpensive heuristics, while achieving accuracy superior to more costly approximation methods. Extensive experiments on long-context understanding benchmarks, across a wide range of models, demonstrate that our method not only outperforms recent competitive baselines in various long-context understanding tasks, but also reduces the eviction cost by up to 14.5×, leading to significantly faster time-to-first-token. Our code is available at `https://github.com/SamsungLabs/LookaheadKV`.

## 1 Introduction

Extending the context length of Large Language Models (LLMs) is becoming increasingly critical for many emerging applications: processing long documents (Bai et al., 2024; Wang et al., 2024; Hsieh et al., 2024), repository-level code understanding and generation (Luo et al., 2024; Liu et al., 2024; Jimenez et al., 2024), in-context learning (Li et al., 2025; Agarwal et al., 2024), etc. However, a central challenge in enabling these applications is that the key-value (KV) cache size grows linearly in sequence length, which rapidly becomes a bottleneck for inference, restricting scalable deployment of such applications. For example, even for moderate-sized models, such as LLaMA3.1–70B (Dubey et al., 2024) in half-precision, storing a single 128K-token sequence already takes up 40GB of memory, while scaling to 1M tokens requires 320GB, exceeding the memory capacity of high-end consumer hardware.

A growing line of work addresses this challenge by identifying salient tokens to achieve effective KV cache eviction without loss of performance (Li et al., 2024; Cai et al., 2024; Galim et al., 2026;

---

[*] Equal contribution [†] Corresponding authors.

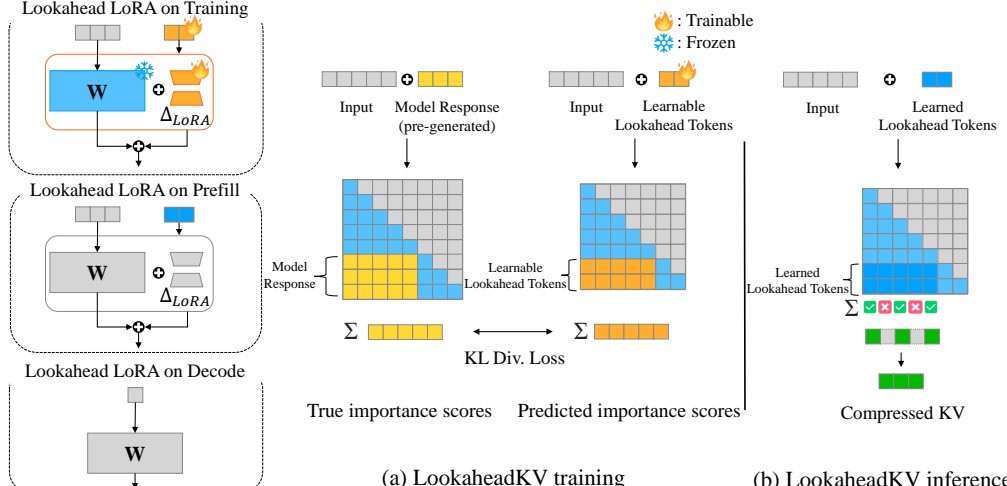

Figure 1: **An overview of LOOKAHEADKV (a) Training.** During training, lookahead tokens and lookahead LoRA are trained to predict the ground-truth importance scores obtained with pre-generated model response via a KL divergence loss. **(b) Inference.** During prefill, LOOKAHEADKV utilizes the learned modules to identify essential tokens and compress the KV cache, facilitating memory-efficient decoding.

Wang et al., 2025; Zhang et al., 2023). Early methods often rely on simple heuristics, in which token importance is estimated based on the self-attention scores of a subset of the input tokens. SnapKV (Li et al., 2024), for instance, leverages the attention weights between the suffix of the input and the preceding context to estimate the importance of each prompt token. More recently, several studies (Galim et al., 2026; Wang et al., 2025) reveal that leveraging the model's response, rather than the input prompt, can greatly improve the eviction quality. Furthermore, they show that a low-cost generated draft response (generated using a smaller draft model (Galim et al., 2026), for instance), which closely approximates the true response, can serve as a powerful proxy for accurately estimating the importance scores.

While these draft-based methods substantially improve eviction quality, they still face a trade-off between performance and latency, since their draft token generation step is computationally expensive. Figure 2 presents the trade-off between accuracy and overhead of different approaches using the QASPER benchmark (Dasigi et al., 2021) and LLaMA3.1-8B-Instruct (Dubey et al., 2024). While simpler approaches like SnapKV induce minimal latency overhead, they suffer severe performance degradation under highly constrained budget settings. On the other hand, Lookahead Q-Cache (LAQ) (Wang et al., 2025), a draft-based approach, shows impressive results even in extremely limited budget settings. However, this approach incurs prohibitive computational overhead by generating an extra draft response, which limits its practicality in latency-sensitive applications such as mobile devices.

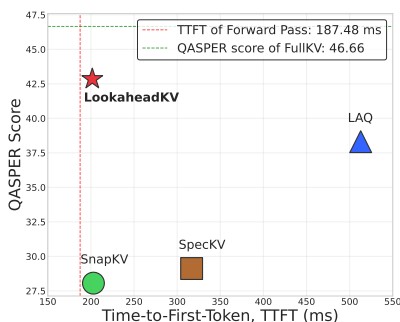

Figure 2: Accuracy-overhead trade-off across KV cache eviction methods.

To overcome this limitation, we introduce LOOKAHEADKV, a novel KV cache eviction method that augments LLMs with parameter-efficient modules, capable of accurately predicting future attention patterns, eliminating the need for costly draft token generation. As shown in Figure 2, our method effectively overcomes the accuracy-overhead trade-off, achieving minimal performance loss with negligible overhead. LOOKAHEADKV, as depicted in Figure 1, employs a set of learnable special tokens, together with lookahead LoRA modules, novel selectively activated low-rank adapters, to produce queries that can reliably estimate token-importance scores. By fine-tuning them to predict

the true importance scores, LOOKAHEADKV effectively minimizes the quality loss incurred by KV cache eviction with marginal inference overhead.

To rigorously assess the effectiveness of LOOKAHEADKV, we evaluate it on a diverse set of long-context benchmarks (Bai et al., 2024; Hsieh et al., 2024; Ye et al., 2025; Zheng et al., 2023) across multiple models of varying sizes (Dubey et al., 2024; Yang et al., 2025). Experimental results consistently demonstrate that LOOKAHEADKV outperforms strong baselines across multiple budgets and context lengths while incurring significantly less eviction latency.

To summarize, our contributions are as follows:

- We propose LOOKAHEADKV, a novel KV cache eviction framework that employs learnable lookahead tokens and special LoRA modules to predict the importance scores from the model's true response without explicitly generating costly approximate response.

- Through extensive experiments, we demonstrate that the proposed approach is effective and robust across different models and context lengths. It remains superior in low-budget settings, providing a useful solution in resource-constrained environments.

- By conducting a rigorous analysis of eviction latency, both theoretically and empirically, we show that our method incurs negligible eviction overhead of less than $2.16\%$ at 32K context length, which is up to $14.5\times$ lower than the overhead incurred by draft-based approaches.

## 2 BACKGROUND

The primary objective of the KV cache eviction methods considered in this work, including our proposed approach, is to accurately estimate the importance score of individual key-value pairs of prompt tokens using attention weights, in order to guide the eviction process. In the following section, we formally define the problem of KV cache eviction and briefly discuss how prior methods have approached it.

**KV Cache Eviction Using Importance Scores.** Let $X = \{x_1, ..., x_{n_{\text{in}}}\}$ be an input token sequence (e.g., a user instruction, part of a code snippet, etc.) and $Y = \{y_1, ..., y_{n_{\text{out}}}\}$ the model's generated response to $X$. For a given layer and attention head in an LLM, the attention scores of the complete sequence are given by:

$$\mathbf{Q} = \begin{bmatrix} \mathbf{X} \\ \mathbf{Y} \end{bmatrix} \mathbf{W}_q \qquad \mathbf{K} = \begin{bmatrix} \mathbf{X} \\ \mathbf{Y} \end{bmatrix} \mathbf{W}_k \qquad \mathbf{A} = \text{Softmax}\left( \frac{\mathbf{Q}\,\mathbf{K}^\top}{\sqrt{d}} \right), \qquad (1)$$

where $\mathbf{X} = [\mathbf{x}_1, ..., \mathbf{x}_{n_{\text{in}}}]^\top \in \mathbb{R}^{n_{\text{in}} \times d}$ and $\mathbf{Y} = [\mathbf{y}_1, ..., \mathbf{y}_{n_{\text{out}}}]^\top \in \mathbb{R}^{n_{\text{out}} \times d}$ are the hidden states of the input prompt and model-generated response, respectively. For better readability, we omit the layer and head index. We define the ground-truth importance scores $\mathbf{s}_{\text{GT}} = [s_1, ..., s_{n_{\text{in}}}]$ of the KV cache as the average cross-attention scores between the queries of $\mathbf{Y}$ and the keys of $\mathbf{X}$, i.e., $s_j = \frac{1}{n_{\text{out}}} \sum_{i=n_{\text{in}}+1}^{n_{\text{in}}+n_{\text{out}}} \mathbf{A}_{i,j}$. Intuitively, these scores quantify the relative contribution of each prompt token's key–value pair to the model's response generation. Based on these scores, the pruned KV cache can be obtained by retaining a subset of (e.g., TopK) important KV pairs to minimize the attention output perturbation, such that $\text{Attn}(x, \text{KV}_{\text{orig}}) \approx \text{Attn}(x, \text{KV}_{\text{GT}})$, where $\text{KV}_{\text{orig}}$ and $\text{KV}_{\text{GT}}$ are the original and evicted KV cache using the ground-truth importance scores, respectively.

However, since the model's true future response is unknown during the prefill phase, such scores cannot be computed directly. Consequently, prior methods resorted to constructing a surrogate response sequence $\tilde{\mathbf{Y}} = [\tilde{y}_1, ..., \tilde{y}_{n_{\text{window}}}]^\top \in \mathbb{R}^{n_{\text{window}} \times d}$ to approximate the model's (partial) future response and predict the attention pattern:

$$\tilde{\mathbf{Q}} = \begin{bmatrix} \mathbf{X} \\ \tilde{\mathbf{Y}} \end{bmatrix} \mathbf{W}_q \qquad \tilde{\mathbf{K}} = \begin{bmatrix} \mathbf{X} \\ \tilde{\mathbf{Y}} \end{bmatrix} \mathbf{W}_k \qquad \tilde{\mathbf{A}} = \text{Softmax}\left( \frac{\tilde{\mathbf{Q}}\,\tilde{\mathbf{K}}^\top}{\sqrt{d}} \right), \qquad (2)$$

resulting in the estimated importance score vector $\mathbf{s}_{\text{approx}} = [\tilde{s}_1, ..., \tilde{s}_{n_{\text{in}}}]$ whose entries are computed as $\tilde{s}_j = \frac{1}{n_{\text{window}}} \sum_{i=n_{\text{in}}+1}^{n_{\text{in}}+n_{\text{window}}} \tilde{\mathbf{A}}_{i,j}$. In short, these methods aim to obtain the estimated score vector whose ranking is similar to that of the ground-truth, such that the overlap between the retained KV

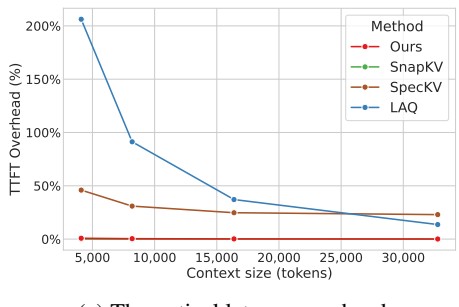
(a) Theoretical latency overhead

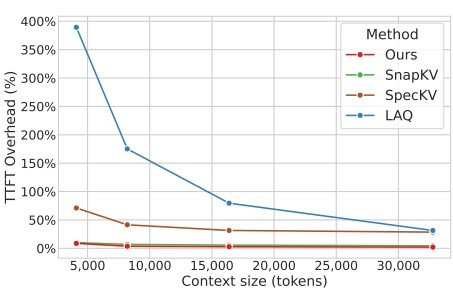
(b) Actual latency overhead

Figure 3: Time-to-First-Token (TTFT) latency overhead ratio across context lengths. Similar to SnapKV, LOOKAHEADKV introduces negligible TTFT overhead across all tested context lengths; draft-based methods (LAQ, SpecKV) incur substantial latency, especially for shorter contexts.

pairs and $KV_{GT}$ is high. Various approaches have been suggested to approximate the future response for effective KV cache eviction.

**Prompt-based Approaches.** SnapKV (Li et al., 2024) uses the suffix of input prompt to compute the estimate of the future importance scores. It has been widely adopted as a simple and effective KV cache eviction method because it can reuse attention weights from the prefill forward pass, requiring only marginal extra computation.

**Draft-based Approaches.** Recently, several works have proposed to use a low-cost generator to generate a (partial) approximate response first, and subsequently use it to estimate the future importance scores. For example, SpecKV (Galim et al., 2026) employs a smaller LLM to generate a draft response, while Lookahead Q-Cache (LAQ) (Wang et al., 2025) first applies SnapKV to the target model to generate a draft response, which is in turn used to approximate the future salience.

These draft-based methods have consistently shown superior performance compared to simple heuristics (Li et al., 2024), demonstrating the effectiveness of employing surrogate future response. However, the explicit draft generation step still incurs substantial additional compute, resulting in significant increase in latency, as shown in Figure 3. In summary, existing methods face a clear trade-off: inexpensive heuristics are fast but less accurate, whereas draft-based techniques improve performance at the cost of increased inference time.

## 3 PROPOSED METHOD: LOOKAHEADKV

To overcome the challenge of fast and accurate importance prediction, we introduce LOOKA-HEADKV, a framework that augments the LLM with a set of lightweight learnable modules which are optimized to predict ground-truth importance scores. LOOKAHEADKV achieves the best of both worlds *by glimpsing into the future without generation:* **1)** it eliminates the need for the explicit draft generation step, resulting in significantly faster KV cache eviction. **2)** it employs learned special tokens that serve as implicit future response for importance estimation, leveraging the strength of draft-based methods without their computational overhead.

### 3.1 MAIN COMPONENTS

**Learnable Lookahead Tokens.** LOOKAHEADKV performs KV cache eviction using a set of learnable special tokens during the prefill phase, followed by auto-regressive decoding with the preserved KV cache. For a given input sequence $X$, our framework appends a sequence of trainable soft lookahead tokens $P = \{p_1, ..., p_{n_{\text{lookahead}}}\}$ whose queries in each attention head are used to estimate the attention pattern of the true model response. In other words, these tokens are trained to compress the attention information of the true response to serve as the "observation window" in the eviction phase. These are initialized randomly and added to the vocabulary before training. Note that the lookahead tokens are used during the prefill stage only for eviction, and introduce no overhead for the decoding stage.

**Lookahead LoRA.** To enhance the quality of estimation, we introduce lookahead LoRA, a novel low-rank adapter module that only activates for the lookahead tokens. Lookahead LoRA provides complementary performance gains by allowing these tokens to learn richer representations, enabling their queries to more accurately predict token importance. The selective activation mechanism of the LoRA modules ensures that the outputs of normal input tokens are unchanged, preserving the original model behavior. Since the original model weights remain unaltered, LOOKAHEADKV modules can be selectively enabled or disabled depending on the particular requirements of a given application, thereby broadening the method's applicability.

Combining the modules together, LOOKAHEADKV computes the queries and keys of the complete sequence as follows:

$$\mathbf{Q}_{\text{LKV}} = \begin{bmatrix} \mathbf{X} \\ \mathbf{P} \end{bmatrix} \mathbf{W}_q + \begin{bmatrix} \mathbf{0} \\ \mathbf{P} \end{bmatrix} \Delta \mathbf{W}_q \qquad \mathbf{K}_{\text{LKV}} = \begin{bmatrix} \mathbf{X} \\ \mathbf{P} \end{bmatrix} \mathbf{W}_k + \begin{bmatrix} \mathbf{0} \\ \mathbf{P} \end{bmatrix} \Delta \mathbf{W}_k, \qquad (3)$$

where $\mathbf{P} \in \mathbb{R}^{n_{\text{lookahead}} \times d}$ denotes the hidden states of the lookahead embeddings, and $\Delta \mathbf{W}_q$, $\Delta \mathbf{W}_k$ are the lookahead LoRA modules for query and key projections. Similar to prior methods (Li et al., 2024; Cai et al., 2024; Zhang et al., 2023), we use the attention matrix $\mathbf{A}_{\text{LKV}} = \text{Softmax}(\frac{\mathbf{Q}_{\text{LKV}} \mathbf{K}_{\text{LKV}}^\top}{\sqrt{d}})$, to estimate the importance score $\tilde{s}_j = \frac{1}{n_{\text{lookahead}}} \sum_{i=n_{\text{in}}+1}^{n_{\text{in}}+n_{\text{lookahead}}} \mathbf{A}_{\text{LKV}\,i,j}$, and retain Top-K KV pairs with the highest importance scores.

## 3.2 LOOKAHEADKV TRAINING

We train LOOKAHEADKV modules to compress the attention pattern of the true future response, using the model-generated responses as target. Given a data pair $(X, Y)$, one iteration of LOOKAHEADKV training consists of the following steps:

1. **GT Forward Pass.** For each layer $l = 1, ..., L$ and head $h = 1, ..., H$, the ground-truth importance scores $\mathbf{s}_{\text{GT}}^{l,h}$ between the input prompt $X$ and model-generated response $Y$ are computed.

2. **Lookahead Forward Pass.** For each layer $l$ and head $h$, we obtain the importance score estimates $\mathbf{s}_{\text{LKV}}^{l,h}$ between the input prompt $X$ and the lookahead tokens $P$.

3. **Loss Computation.** We first normalize all score vectors such that they sum to 1, and compute the average KL divergence loss between the GT and LOOKAHEADKV importance scores across all heads and layers:

$$\mathcal{L}_{\text{LKV}} = \frac{1}{L \cdot H} \sum_{l}^{L} \sum_{h}^{H} \mathrm{D}_{\text{KL}}\big( \hat{\mathbf{s}}_{\text{GT}}^{l,h} \parallel \hat{\mathbf{s}}_{\text{LKV}}^{l,h} \big). \qquad (4)$$

where $\hat{\mathbf{s}}$ is the $L_1$-normalized importance scores such that $\hat{\mathbf{s}} = \mathbf{s}/\|\mathbf{s}\|_1$. The loss is backpropagated to update the both the lookahead embeddings and LoRA modules, while all other LLM layers remain frozen, as shown in Figure 1. The pseudo-code for LOOKAHEADKV training and eviction is given in Algorithm 1 and Algorithm 2.

**Training Objective.** Inspired from works on distilling attention scores (Wang et al., 2020; Izacard & Grave, 2021), we minimize the KL divergence between these normalized attention scores. As our attentions scores are normalized, this KL divergence is equivalent to the popular ListNet (Cao et al., 2007) ranking loss, with $\phi$ of ListNet as identity instead of $\exp$.

**Lookahead LoRA Overhead.** In principle, one can apply lookahead LoRA to any subset of the linear layers to tradeoff accuracy and latency. However, even when lookahead LoRA is applied to every linear layer, there is a minor increase ($<1.3\%$) in latency compared to not using lookahead LoRA at all (see Table 5 for ablation results), while significantly boosting performance compared to not using LoRA. Consequently, we train LOOKAHEADKV with LoRA modules applied to all linear layers.

To avoid materializing the full attention score matrix, we use FlashAttention (Dao et al., 2022) in the forward pass, coupled with eager attention for importance score computation and loss backpropagation, as detailed in Section C.

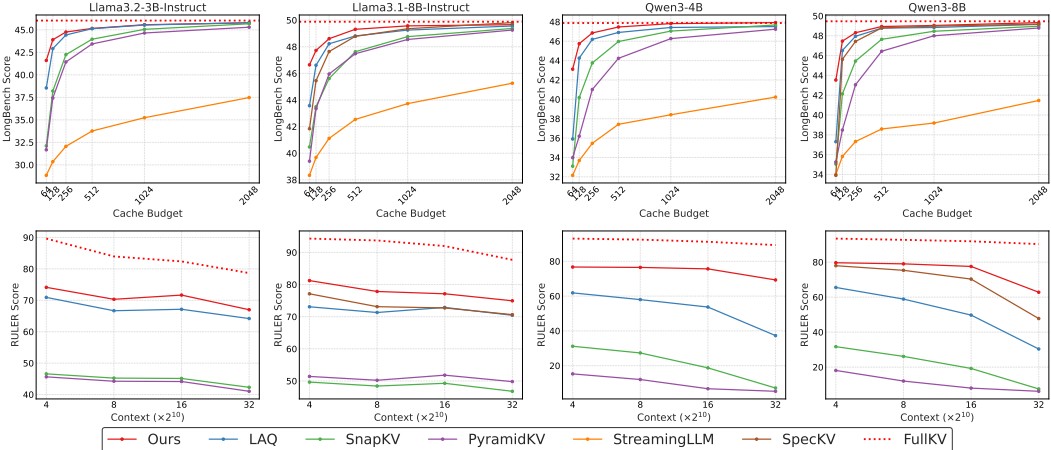

Figure 4: Top row: Average LongBench results across multiple budgets and models. Bottom row: Average RULER results across varying context lengths with a fixed budget of 128. Across all tested models, budgets and context lengths, LOOKAHEADKV consistently demonstrates superior performance.

## 4 EXPERIMENTS

### 4.1 TRAINING

**Dataset.** To encourage the model to learn from diverse attention patterns, we curate training samples of varying lengths and sources, comprising both instruction-following datasets as well as pretraining texts. We collect 50K samples from the long_sft subset of the ChatQA2 (Xu et al., 2025) dataset, 20K samples from the Tulu (Lambert et al., 2025) instruction-following dataset, 7K samples from the Stack (Kocetkov et al., 2023), and 9K few-shot completion data samples that we create based on the training splits of the MetaMath, ARC, and HellaSwag datasets, originally curated in Pal et al. (2024). For instruction-following data, we remove the last assistant response and use the target model to obtain the $(X, Y)$ pairs of input prompt and model response. For pretraining documents, we first truncate the text at random positions to obtain $X$, and use the target model to complete the sequence to obtain $Y$. We limit the maximum input sequence length to 16K, and generate all training responses using greedy decoding and max generation length of 512.

**Training Details.** We apply LOOKAHEADKV on two widely used open-source architectures, LLaMA (Dubey et al., 2024) and Qwen (Yang et al., 2025), covering three model sizes each: LLaMA3.2-1B, LLaMA3.2-3B, LLaMA3.1-8B, Qwen3-1.7B, Qwen3-4B, and Qwen3-8B. For all models, we set the lookahead size $n_{\text{lookahead}} = 32$, and apply LoRA to all projection and feed-forward modules ($\mathbf{W}_q, \mathbf{W}_k, \mathbf{W}_v, \mathbf{W}_o, \mathbf{W}_{up}, \mathbf{W}_{down}$, and $\mathbf{W}_{gate}$) with rank $r = 8$ and scaling factor $\alpha = 32$. This configuration introduces less than 0.5% additional trainable parameters across all models, as summarized in Table 1. Full hyperparameter settings are provided in Table 16.

Table 1: Additional trainable parameters introduced by LOOKA-HEADKV.

| Model | Trainable Params | |
|---|---|---|
| | Params | % of Model |
| LLaMA3.2-1B | 5.4M | 0.44 |
| LLaMA3.2-3B | 11.9M | 0.37 |
| LLaMA3.1-8B | 20.6M | 0.26 |
| Qwen3-1.7B | 8.5M | 0.49 |
| Qwen3-4B | 16.2M | 0.40 |
| Qwen3-8B | 21.5M | 0.26 |

### 4.2 EVALUATION SETUP

We evaluate our method on a comprehensive suite of benchmarks: LongBench (Bai et al., 2024), RULER (Hsieh et al., 2024), LongProc (Ye et al., 2025), and MT-Bench (Zheng et al., 2023). Long-Bench is a multi-task benchmark that assesses the long-context understanding capability across diverse tasks, such as question answering, summarization, and code completion. We report results on the 16 English tasks, and use the average score as the main metric. RULER is another multi-task synthetic benchmark, primarily comprising 13 Needle-in-a-Haystack-style subtasks. Each sample can be constructed at varying sequence lengths, allowing systematic evaluation of scaling behav-

Table 2: MT-Bench evaluation results. Bold and underlined values indicate best and second best scores, respectively.

| Model | Budget | PyramidKV | SnapKV | StreamingLLM | SpecKV | LAQ | LOOKAHEADKV |
|---|---|---|---|---|---|---|---|
| | | *FullKV score: 5.72* | | | | | |
| LLaMA3.2-1B | 64 | 4.64 | 4.70 | 4.54 | N/A | 5.03 | **5.21** |
| | 128 | 5.10 | 5.39 | 4.94 | N/A | 5.45 | **5.60** |
| | 256 | 5.49 | **5.67** | 5.37 | N/A | 5.64 | 5.62 |
| | | *FullKV score: 7.35* | | | | | |
| LLaMA3.2-3B | 64 | 6.30 | 6.28 | 5.96 | 6.52 | 6.48 | **6.87** |
| | 128 | 6.93 | 7.03 | 6.42 | 7.02 | 6.93 | **7.26** |
| | 256 | 7.19 | 7.30 | 7.20 | 7.28 | **7.43** | 7.30 |
| | | *FullKV score: 7.77* | | | | | |
| LLaMA3.1-8B | 64 | 6.85 | 6.80 | 6.17 | 6.77 | 7.1 | **7.26** |
| | 128 | 7.39 | 7.50 | 6.84 | 7.34 | 7.54 | **7.63** |
| | 256 | 7.76 | 7.72 | 7.41 | 7.84 | 7.72 | **7.92** |
| | | *FullKV score: 7.19* | | | | | |
| Qwen3-1.7B | 64 | 5.81 | 5.95 | 5.83 | N/A | 6.19 | **6.70** |
| | 128 | 6.38 | 6.65 | 6.16 | N/A | 6.91 | **7.12** |
| | 256 | 6.90 | 6.94 | 6.91 | N/A | 7.02 | **7.20** |
| | | *FullKV score: 8.02* | | | | | |
| Qwen3-4B | 64 | 6.85 | 6.60 | 6.24 | 7.05 | 7.06 | **7.69** |
| | 128 | 7.55 | 7.71 | 7.24 | 7.78 | 7.70 | **8.12** |
| | 256 | 7.90 | **8.20** | 7.87 | 8.11 | 8.12 | 8.06 |
| | | *FullKV score: 8.48* | | | | | |
| Qwen3-8B | 64 | 7.33 | 7.26 | 6.81 | 7.69 | 7.58 | **8.04** |
| | 128 | 7.85 | 7.94 | 7.64 | 7.97 | 8.24 | **8.41** |
| | 256 | 8.42 | 8.43 | 8.34 | 8.45 | **8.56** | 8.51 |

ior. Similar to LongBench, we use average score as the main metric, and report the results at 4K, 8K, 16K and 32K context lengths. We further evaluate the model's long-form output generation capability on the HTML to TSV task from LongProc, which involves converting structured information from long HTML documents into TSV format. Finally, MT-bench provides a comprehensive multi-turn question set, spanning various domains such as writing, coding, and math.

**Baselines.** We evaluate our method against popular KV cache eviction methods: **1) SnapKV** (Li et al., 2024), **2) PyramidKV** (Cai et al., 2024), and **3) StreamingLLM** (Xiao et al., 2024). We also compare our approach to stronger, more recent baselines that require costly approximate future response generation, such as **4) Lookahead Q-Cache** (LAQ) (Wang et al., 2025), and for 8B-scale models, **5) SpecKV** (Galim et al., 2026). In all experiments, Llama3.2-1B-Instruct and Qwen3-1.7B are used as draft models for Llama3.1-8B-Instruct and Qwen3-8B, respectively. We follow the standard eviction configuration settings for all baseline methods, which we detail in Section F.

## 4.3 PERFORMANCE RESULTS

**LongBench Evaluation.** Figure 4 shows the average LongBench scores of LOOKAHEADKV and baselines, across cache budget settings ranging from 64 to 2048. Our method consistently demonstrates superior performance across all models and all budgets tested, demonstrating the effectiveness and robustness of our approach. Overall, results show that expensive draft-based methods (e.g., LAQ and SpecKV) outperform simple baselines, corroborating that employing approximate future response for importance estimation is effective. Nevertheless, our method significantly outperforms draft-based approaches, especially at lower budget settings, highlighting that learning to estimate future importance is crucial for performance preservation. Due to space limitations, we report the results of 1B-scale models in Section E.

**RULER Evaluation.** We report the RULER evaluation results of all methods with a fixed budget of 128 in Figure 4 (1B-scale results are provided in Section E). LOOKAHEADKV consistently outperforms other baseline approaches here as well, maintaining strong performance across all evaluated context lengths. Despite being trained on a maximum sequence length of 16K, LOOKAHEADKV effectively generalizes to a longer context length of 32K. We conduct additional experiments on the impact of training context length in Section 5.4.

**Long-form Output Evaluation.** To further validate LOOKAHEADKV's ability to generate long-form outputs, we evaluate our method on the HTML to TSV task from LongProc. We assess

Table 3: Theoretical and empirical cost analysis of LLaMA3.1-8B at $C = 128$.

| Context Length | Method | Theoretical Cost | | | | Empirical Cost | |
|---|---|---|---|---|---|---|---|
| | | Compute (TFLOPs) | Memory Traffic (GB) | TTFT (ms) | TTFT Overhead (ms) | TTFT (ms) | TTFT Overhead (ms) |
| 8K | Forward Pass Only | 136 | 13 | 257 | N/A | 291 | N/A |
| | LOOKAHEADKV | 137 | 13 | 258 | 1.03 | 302 | 11 |
| | SnapKV | 136 | 13 | 257 | 0.01 | 311 | 20 |
| | SpecKV | 159 | 81 | 337 | 79.53 | 411 | 121 |
| | LAQ | 137 | 445 | 492 | 234.59 | 800 | 509 |
| 32K | Forward Pass Only | 928 | 13 | 1754 | N/A | 1760 | N/A |
| | LOOKAHEADKV | 929 | 13 | 1755 | 1.74 | 1798 | 38 |
| | SnapKV | 928 | 13 | 1754 | 0.01 | 1838 | 78 |
| | SpecKV | 1115 | 106 | 2156 | 402.80 | 2263 | 503 |
| | LAQ | 930 | 451 | 1993 | 239.26 | 2314 | 554 |

LOOKAHEADKV and baseline methods under two input–output settings: 12K–0.5K and 23K–2K tokens, both at a fixed cache budget ratio of 30%.

Figure 5 presents the results on the HTML to TSV task using LLaMA-3.1-8B. Across both sequence-length configurations, LOOKA-HEADKV consistently outperforms prior approaches. We hypothesize that LOOKAHEADKV, learning to predict the attention pattern of the entire future response, is particularly superior in long-form generation tasks compared to draft-based methods that rely only on partial future response as the observation window.

**Multi-turn Evaluation.** To test our method under multi-turn conversation setting, we evaluate LOOKAHEADKV and baselines on MT-Bench (Zheng et al., 2023). The generated responses are evaluated using Qwen3-235B-A22B as the LLM judge. The results in Table 2 indicate

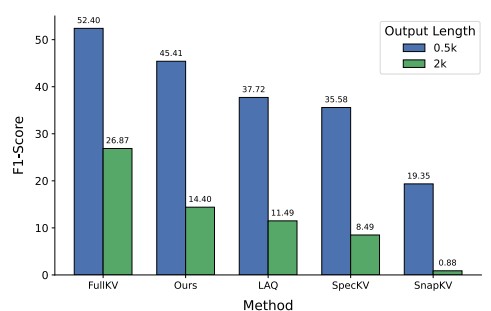

Figure 5: HTML-to-TSV evaluation results using LLaMA3.1-8B.

that LOOKAHEADKV is either on par or superior across all models and budgets tested. LOOKA-HEADKV is particularly robust in lower budget settings (e.g., $C = [64, 128]$), where it consistently outperforms all other methods.

## 5  ANALYSIS

### 5.1  EFFICIENCY COMPARISON

Efficiency is assessed by measuring the Time-To-First-Token (TTFT) for LLaMA-3.1-8B across multiple context lengths. While we utilize official codebases for most baselines, LAQ was re-implemented due to the absence of an official release. Because the latency of a method can vary significantly depending on the implementation, we conduct rigorous analysis and derive the theoretical latency for each method, based on the analytical model proposed in Davies et al. (2025). We discuss further details in Section B.

Table 3 presents the results of the TTFT analysis for 8K and 32K context lengths (see Table 15 for 4K and 16K results). Overall, we observe that draft-based methods incur significant overhead, either due to increased computation (SpecKV) or memory traffic (LAQ). On the contrary, LOOKAHEADKV requires marginal additional cost across all tested context lengths, reducing eviction overhead by 14.5× compared to LAQ at 32K sequence length.

Table 4: Average LongBench performance at different temperature settings on LLaMA3.1-8B, with $C = 128$. LOOKAHEADKV outperforms baselines across all tested temperature settings.

| Method | FullKV | SnapKV | SpecKV | LAQ | LOOKAHEADKV |
|---|---|---|---|---|---|
| Greedy | 49.88 | 43.50 | 45.45 | 46.61 | **47.72** |
| $T = 0.2$ | 49.58 (-0.60%) | 43.29 (-0.48%) | 44.99 (-1.01%) | 46.73 (+0.26%) | **47.75** (+0.06%) |
| $T = 0.8$ | 47.82 (-4.13%) | 41.39 (-4.85%) | 43.43 (-4.44%) | 45.27 (-2.87%) | **45.81** (-4.00%) |

Table 5: 2D ablation across lookahead sizes and trainable modules, on LLaMA3.2-1B. Average LongBench scores with cache budget of 64 and TTFT overhead are reported.

| Module | $n_{\text{lookahead}} = 4$ | | $n_{\text{lookahead}} = 8$ | | $n_{\text{lookahead}} = 16$ | | $n_{\text{lookahead}} = 32$ | | $n_{\text{lookahead}} = 64$ | | $n_{\text{lookahead}} = 128$ | |
|---|---|---|---|---|---|---|---|---|---|---|---|---|
| | score | overhead(%) | score | overhead(%) | score | overhead(%) | score | overhead(%) | score | overhead(%) | score | overhead(%) |
| emb-only | 25.5 | 3.4 | 25.7 | 3.8 | 26.4 | 3.4 | 26.4 | 4.2 | 25.8 | 7.3 | 26.2 | 10.7 |
| QV | 26.5 | 3.7 | 26.4 | 4.1 | 26.9 | 4.0 | 26.9 | 4.4 | 26.7 | 7.7 | 27.1 | 10.7 |
| all | 26.6 | 4.2 | 27.0 | 4.2 | 27.0 | 4.7 | 27.1 | 5.0 | 27.1 | 8.5 | 27.0 | 11.0 |

## 5.2 EFFECT OF STOCHASTIC DECODING

We analyze the effect of stochastic generation on LOOKAHEADKV's performance by evaluating our method using two temperature settings: $[0.2, 0.8]$. Results in Table 4 show that LOOKAHEADKV maintains superior performance over all other baselines across all temperature settings. Further, we observe that performance degradation at high temperature setting (3-4% at $T = 0.8$) is consistent across all methods, including FullKV, indicating that stochasticity in inference affects all approaches similarly. We further discuss the interplay between stochastic decoding for training data generation and LOOKAHEADKV performance in Section E.3

## 5.3 ABLATION ON TRAINABLE MODULES

We study the impact of lookahead size $n_{\text{lookahead}}$ and LoRA placement through a 2D ablation across six lookahead sizes (4, 8, 16, 32, 64, 128) and three configurations: emb-only (No LoRA applied), QV (LoRA applied to Q and V), and all (LoRA applied to all linear layers). The results indicate that both larger lookahead windows and broader LoRA coverage generally improve average LongBench performance. However, performance gains saturate at $n_{\text{lookahead}} = 32$; increasing the lookahead size beyond this point yields diminishing returns while incurring a noticeable increase in inference overhead. On the other hand, applying lookahead LoRA to all layers results in relatively minor rise in TTFT while significantly improving the performance across all lookahead sizes. Based on this analysis, we set $n_{\text{lookahead}} = 32$ and apply LoRA to all linear modules in our main experiments.

## 5.4 ROBUSTNESS TO TRAINING CONTEXT LENGTH

Transformer-based language models trained with fixed context lengths often struggle to generalize beyond their training window. Similarly, one may raise concern about the context length generalization of our method. To examine this effect, we apply LOOKAHEADKV training to LLaMA-3B with limited training context lengths of 2K, 4K, and 8K, and evaluate on RULER (Figure 6). We observe that while longer training context length yields better performance as expected, training on shorter contexts still remains effective with relatively minor degradation in performance, demonstrating that our method generalizes robustly to unseen sequence lengths.

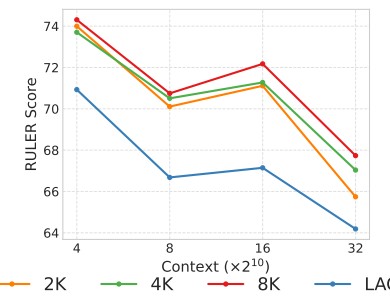

Figure 6: RULER evaluation on LOOKAHEADKV trained with shorter contexts.

## 6 RELATED WORK

**KV Cache Eviction.** Early analyses revealed that attention scores tend to be sparse (Zhang et al., 2023), implying that only a small subset of KV entries substantially contributes to the attention out-

put. Further, subsequent work showed that the importance of these tokens remains stable throughout generation, i.e., tokens deemed important early on tend to stay important (Liu et al., 2023). These observations motivated a range of eviction methods aimed at discarding unimportant KV entries while preserving model performance. Representative methods such as H2O (Zhang et al., 2023), NACL (Chen et al., 2024) and TOVA (Oren et al., 2024) rely on attention scores to estimate token importance and evict low-importance KV pairs. In contrast, other works propose to use alternative importance metrics for eviction (Park et al., 2025; Guo et al., 2024; Geng et al., 2025) to improve importance estimation or address the challenge of materializing the full attention matrix.

**Prefill KV Cache Eviction.** A specific line of work, which we discuss extensively in our paper, focuses on eviction of prefill KV cache. SnapKV (Li et al., 2024) introduced the notion of an "observation window" consisting of the suffix of the input prompt, which is used to predict important tokens to keep for subsequent response generation. SpecKV (Galim et al., 2026) proposed to generate an approximate response with a smaller model and use the resulting tokens as a more reliable observation window for future importance prediction. Lookahead Q-Cache (Wang et al., 2025) first applies a simple eviction method, such as SnapKV, to obtain a partial low-cost draft response, then re-evicts KV entries based on the importance scores derived from the draft. KVzip (Kim et al., 2025) adopts a query-agnostic strategy by inserting a repeated prompt and measuring which KV entries are essential for accurately reconstructing the input. Orthogonal to these approaches, several works proposed to allocate non-uniform budgets for each layer (Cai et al., 2024) and head (Feng et al., 2024) to further improve performance.

**Prompt Tuning for Task Adaptation.** Another line of work closely related to ours is parameter-efficient finetuning through learned prompts. Prompt Tuning (Lester et al., 2021) inserts a sequence of continuous, learnable embeddings into the frozen LLM for downstream task adaptation, while Prefix-Tuning (Li & Liang, 2021) extends this idea by pre-pending learned vectors across multiple layers. Further, P-Tuning v2 (Liu et al., 2022) demonstrated that prompt-based adaptation scales well across a wide range of model sizes. Unlike conventional prompt-tuning methods that aim to improve task performance, our work leverages learned prompts to predict internal model statistics, thereby enhancing computational efficiency rather than accuracy.

Training objectives similar to ours have been used in distillation (Wang et al., 2020), or in ranking/retrieval (Cao et al., 2007; Izacard & Grave, 2021). Some contemporaneous works (Greenewald et al., 2025; Peng et al., 2025; Samragh et al., 2025) also propose LoRA modules that selectively activate only for some tokens.

# 7 CONCLUSION AND LIMITATION

We introduce LOOKAHEADKV, a trainable KV cache eviction framework that accurately predicts token importance without relying on explicit draft generation. The method augments a frozen LLM with a small set of learnable lookahead tokens and lookahead LoRA modules that activate only on these tokens. Trained to match ground-truth importance distributions across layers and heads, LOOKAHEADKV achieves performance superior to costly draft-based approaches. Across a wide range of model families and long-context benchmarks, our approach consistently outperforms prior methods, especially in low-budget regimes. It introduces less than 0.5% additional parameters and incurs only a marginal increase in prefill latency.

Due to limited compute resources, we were unable to conduct experiments on larger-sized models. However, experimental results indicate that LOOKAHEADKV improves both performance and latency of KV cache eviction across a variety of model sizes. LOOKAHEADKV currently focuses on the prefill KV cache eviction; extending LOOKAHEADKV to also perform decoding-stage eviction remains an interesting future work.

ACKNOWLEDGMENT

We would like to thank Daehyun Kim, Ph.D., Hyeonmok Ko, Ph.D., Ho-young Kim, Anshumann, Mohd Abbas Zaidi, and Harshith Goka for their helpful discussions and generous support in this work.

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

## A  PSEUDO-CODE

The pseudocode for LOOKAHEADKV training and eviction is described in Algorithm 1 and Algorithm 2, respectively.

---

**Algorithm 1** LOOKAHEADKV Training

---

**Require:** dataset $\mathcal{D}$ of input-response pairs
 1: `scores` ← []                ▷ GT importance scores
 2: `estimates` ← []         ▷ score estimates using LOOKAHEADKV
 3: **for** each training sample $(X, Y)$ in dataset $D$ **do**
 4:      **for** each layer $l$ **do**                    ▷ GT pass
 5:          **for** each head $h$ in layer $l$ **do**
 6:              $S \leftarrow$ GT importance score for head $(l, h)$
 7:              `scores.append(`$S$`)`
 8:          **end for**
 9:      **end for**
10:      **for** each layer $l$ **do**                  ▷ lookahead pass
11:          **for** each head $h$ in layer $l$ **do**
12:              $\hat{S} \leftarrow$ importance scores using lookahead embeddings for head $(l, h)$
13:              `estimates.append(`$\hat{S}$`)`
14:          **end for**
15:      **end for**
16:      $L \leftarrow 0$                      ▷ compute loss
17:      **for all** $(S, \hat{S})$ in `scores, estimates` **do**
18:          $L \leftarrow L + \mathrm{D_{KL}}\left( \frac{S}{\|S\|_1} \,\|\, \frac{\hat{S}}{\|\hat{S}\|_1} \right)$
19:      **end for**
20:      $L \leftarrow \frac{L}{|\texttt{scores}|}$
21:      $L.$backward()
22: **end for**

---

**Algorithm 2** LOOKAHEADKV Eviction

---

**Require:** Input prompt $X = \{x_1, \ldots, x_{n_{\mathrm{in}}}\}$
**Require:** cache budget $k$
 1: Append learned lookahead tokens to input and compute the sequence embeddings $\hat{\mathbf{X}} = [\mathbf{X}\ \mathbf{P}]^{\top}$
      ▷ shape: $(n_{\mathrm{in}} + n_{\mathrm{lookahead}}) \times d$
 2: Perform a prefill forward pass with $\hat{\mathbf{X}}$:
 3: **for** each layer $l$ **do**
 4:      **for** each head $h$ **do**
 5:          $\mathbf{A} \leftarrow \mathrm{Softmax}\left( \frac{QK^{\top}}{\sqrt{d}} \right)$         ▷ shape: $(n_{\mathrm{in}} + n_{\mathrm{lookahead}}) \times (n_{\mathrm{in}} + n_{\mathrm{lookahead}})$
 6:          $\hat{\mathbf{A}} \leftarrow \mathbf{A}[n_{\mathrm{in}}:, :n_{\mathrm{in}}]$      ▷ attention between lookahead tokens and input prompt
 7:          $\mathbf{s} \leftarrow \mathrm{MeanReduce}(\hat{\mathbf{A}})$
 8:          $\mathbf{s} \leftarrow \mathrm{Pooling}(\mathbf{s})$          ▷ score vector, shape: $1 \times n_{\mathrm{in}}$
 9:          $\mathcal{I} \leftarrow \mathrm{TopK}(\mathbf{s}, k)$
10:          $K^{\mathrm{kept}} \leftarrow K[\mathcal{I}]$
11:          $V^{\mathrm{kept}} \leftarrow V[\mathcal{I}]$
12:          Cache $(K^{\mathrm{kept}}, V^{\mathrm{kept}})$          ▷ evict unimportant KV pairs
13:          Compute attention output for MLP layer
14:      **end for**
15:      Compute MLP output for next layer
16: **end for**
17: **return**

---

## B    THEORETICAL ESTIMATION DETAILS

This section details our methodology for the theoretical estimation of the Time-to-First-Token (TTFT) latency for various KV cache eviction algorithms. Our analysis is based on the analytical model for FLOPs and memory traffic proposed by Davies et al. (2025). To align configurations of theoretical estimates with those of actual measurements, we simulate the execution of LLaMA3.1-8B on a single NVIDIA H100 80GB GPU with a batch size of 1, assuming all weights and activations are in half-precision. We set KV cache budget size of 128, lookahead size as 32, and window size as 32. We only consider tensor operations which are dominant parts of the computations. To provide estimates that closely reflect real-world performance, our calculations incorporate practical hardware utilization by assuming a flops efficiency of 0.7 and a memory efficiency of 0.9, as described in Li (2023).

To isolate the specific overhead introduced by each eviction algorithm, we first establish a baseline by calculating the theoretical latency of a single forward pass. The TTFT overhead for each eviction method is then determined by subtracting this baseline forward pass latency from the method's total estimated TTFT. For LAQ, the total latency is calculated by summing the costs of its three consecutive steps—the first eviction, low-cost generation of pseudo response, and the second eviction. Similarly, the total latency of SpecKV is estimated by aggregating the latencies of its draft prefill, draft decode, and target model eviction phases. A comprehensive implementation of the code to derive theoretical estimates of all baselines is available in the Supplementary Materials.

## C    IMPLEMENTATION OPTIMIZATION

Efficient attention implementations such as FlashAttention (Dao et al., 2022) do not materialize the full attention score matrix, but is required in our setting to compute importance scores and enable gradient backpropagation. A possible solution is to compute the complete attention matrix using native PyTorch (i.e., eager attention), but this quickly leads to an out-of-memory error as the matrix size grows quadratically with the sequence length, which is incompatible with our training setting (up to 16K sequence length). Fortunately, for our objective, we only require the cross-attention scores between the generated response and the entire input sequence, and the response length is typically much shorter than the input prompt.

Leveraging this observation, we adopt the following approach: for the attention layer's forward pass, we use flash attention, while for importance score computation and backpropagation, we employ eager attention. This reduces the memory requirement of eager attention from $\mathcal{O}((|X| + |Y|)^2)$ to $\mathcal{O}(|X| \cdot |Y| + |Y|^2)$, where $|X|$ and $|Y|$ denote the lengths of the input prompt and model response, respectively, with $|X| \gg |Y|$.

# D   NEED FOR DATA GENERATION

One of the requirements of LOOKAHEADKV training is that the target model's generated responses must be available as training data. However, generating these responses from the model can sometimes be costly, e.g., when applying LOOKAHEADKV across multiple models. Hence, to assess whether this requirement of can be relaxed, we evaluate an alternative setting where training uses the responses from the source datasets instead of model-generated outputs.

We observe in Figure 7 that this substitution leads to a relatively minor drop in average LongBench performance in lower-budget regimes. We hypothesize that if the attention distribution of the model-generated responses and that of the source dataset responses are moderately similar, our method can still successfully learn to accurately predict the importance scores. Overall, these results suggest that, in scenarios where training data generation is impractical, using source responses provides a viable and effective alternative.

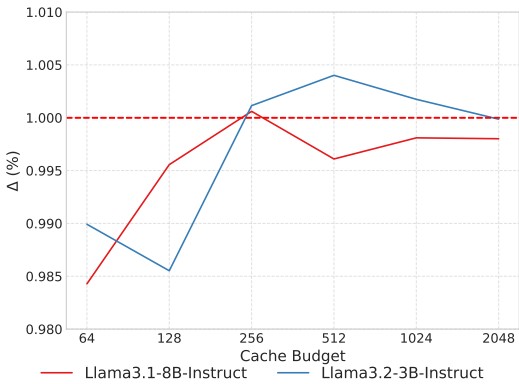

Figure 7: Performance ratio of training using model-generated data vs. source data.

# E  ADDITIONAL RESULTS

In this section, we provide additional experimental results excluded from the main text due to page limitations.

## E.1  RULER EVALUATION ON LONGER CONTEXTS

To explore the capability of LOOKAHEADKV on longer contexts, we evaluate our method on RULER at 64K and 128K context lengths using LLaMA3.1-8B-Instruct with a cache budget of 128. We randomly sample 50 examples per task from the RULER benchmark. As shown in Table 6, LOOKAHEADKV achieves the best performance at these context lengths as well, showing that the effectiveness of our method scales to even longer context lengths.

Table 6: RULER evaluation results on longer context lengths using Llama3.1-8B-Instruct at $C = 128$.

| Context Length | FullKV | LOOKAHEADKV | SnapKV | SpecKV | LAQ |
|---|---|---|---|---|---|
| 64K | 84.15 | **69.45** | 39.64 | 64.02 | 64.10 |
| 128K | 73.72 | **54.83** | 30.56 | 52.62 | 50.67 |

## E.2  EFFECT OF COMBINING SUFFIX WINDOW

To test the effect of incorporating suffix window, as proposed in SnapKV (Li et al., 2024), we augment LOOKAHEADKV by also including queries of the last 32 prompt tokens for importance score estimation. As shown in Table 7, we observe a slight drop in performance when SnapKV importance scores are included. The degraded performance when averaging LOOKAHEADKV importance scores with SnapKV scores, compared to using LOOKAHEADKV scores alone, indicates that the importance predicted by our method is superior to SnapKV.

Table 7: Average LongBench scores using LOOKAHEADKV window only and LOOKAHEADKV + SnapKV-style suffix window, evaluated using LLaMA3.2-1B-Instruct with $C = 64$.

| FullKV | LOOKAHEADKV | LOOKAHEADKV + Suffix Window |
|---|---|---|
| 32.01 | **29.10** | 28.52 (-1.99%) |

## E.3  DISCUSSION OF GENERATION STOCHASTICITY IN LOOKAHEADKV TRAINING

For LOOKAHEADKV training, various stochastic decoding methods may be employed to generate training data. One may hypothesize that the attention matrices induced by responses generated with higher stochasticity may diverge significantly from those from greedy responses, potentially limiting the generalizability of LOOKAHEADKV modules trained exclusively on greedy responses to stochastic inference scenarios. To investigate this, we quantify the similarity between importance score vectors induced by greedy responses and those generated under varying temperature settings.

Table 8 presents recall@512 and Kendall rank correlation coefficients comparing importance scores induced by greedy decoding against stochastic decoding at multiple temperatures using LLaMA3.1-8B. The scores are averaged over 30 randomly selected samples from our training data, across all layers and heads. Even at relatively high temperature ($T = 0.8$), we observe strong persistence of attention patterns. Notably, the deviation is smaller than that induced by responses of a speculative model (Llama3.2-1B, equivalent to the SpecKV setting). This indicates that the ground-truth importance scores derived from stochastically generated responses are highly similar to those from greedy responses, which in turn indicates that greedy-generated training data provides sufficiently robust learning signals for stochastic settings.

Table 8: Importance score similarity with stochastic response using various temperatures vs. greedy response on LLaMA3.1-8B. LLaMA3.2-1B presents the similarity of importance scores using greedy response generated with LLaMA3.2-1B vs. LLaMA3.1-8B.

| Generation Method | $T = 0.2$ | $T = 0.4$ | $T = 0.6$ | $T = 0.8$ | LLaMA3.2-1B |
|---|---|---|---|---|---|
| Recall@512 (%) | 95.06 | 93.73 | 91.40 | 91.37 | 88.66 |
| Kendall's Tau | 91.44 | 88.63 | 84.61 | 84.79 | 80.05 |

## E.4 RESULTS ON LONGBENCH

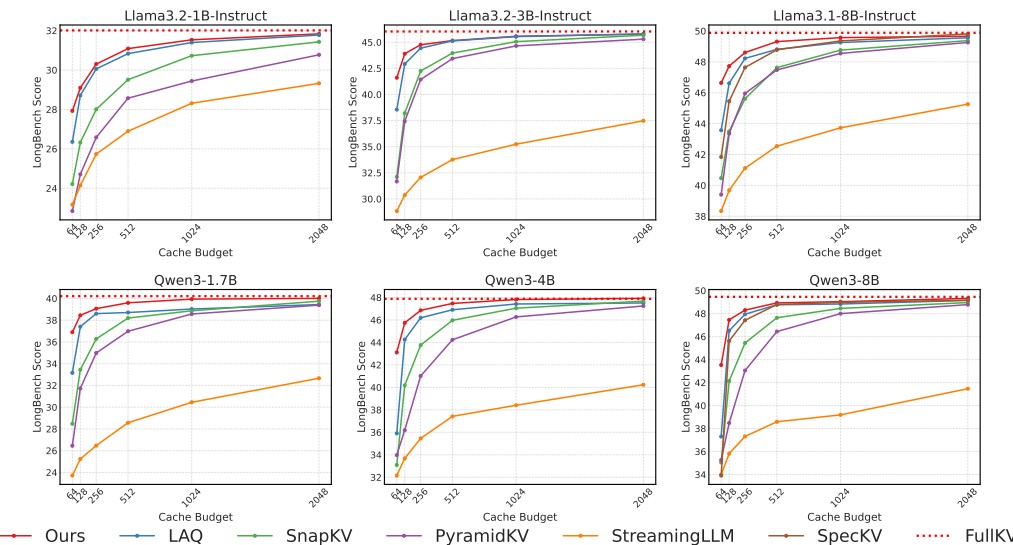

Figure 8: Full Longbench results across multiple cache budgets. 1B-scale results are included.

Table 9: LongBench evaluation results for Llama3.2-1B

| | Single-Document QA | | | Multi-Document QA | | | Summarization | | | Few-shot Learning | | | Synthetic | | Code | | Avg. |
|---|---|---|---|---|---|---|---|---|---|---|---|---|---|---|---|---|---|
| | NrtQA | Qasper | MF-en | HotpotQA | 2WikiMQA | Musique | GovReport | QMSum | MultiNews | TREC | TriviaQA | SAMSum | PCount | Pre | Lcc | RB-P | |
| FullKV | 19.24 | 15.96 | 42.47 | 35.53 | 29.42 | 19.87 | 28.34 | 22.18 | 25.64 | 64.00 | 80.85 | 38.83 | 3.00 | 4.78 | 38.63 | 43.35 | 32.01 |
| *KV Cache Size = 64* | | | | | | | | | | | | | | | | | |
| StreamingLLM | 14.42 | 12.30 | 24.02 | 24.04 | 24.05 | 9.35 | 13.56 | 19.23 | 13.61 | 35.50 | 68.94 | 29.09 | 4.00 | 3.71 | 37.61 | 37.33 | 23.17 |
| SnapKV | 14.24 | 12.18 | 30.53 | 27.30 | 25.66 | 12.44 | 14.27 | 19.27 | 12.37 | 35.00 | 73.57 | 29.29 | 2.00 | 4.49 | 38.01 | 36.84 | 24.22 |
| PyramidKV | 13.33 | 11.36 | 26.05 | 25.34 | 24.01 | 10.57 | 13.64 | 19.49 | 11.96 | 35.00 | 69.95 | 27.75 | 1.50 | 4.33 | 35.25 | 36.01 | 22.85 |
| LAQ | 17.21 | 12.76 | 37.30 | 30.27 | 27.36 | 14.22 | 16.42 | 20.09 | 14.28 | 39.50 | 76.02 | 31.19 | 3.00 | 4.58 | 39.15 | 38.37 | 26.36 |
| LOOKAHEADKV | 17.69 | 13.30 | 40.80 | 33.66 | 29.80 | 16.95 | 18.76 | 20.65 | 18.97 | 45.50 | 80.12 | 34.77 | 2.50 | 3.22 | 33.69 | 36.47 | **27.93** |
| *KV Cache Size = 128* | | | | | | | | | | | | | | | | | |
| StreamingLLM | 14.84 | 12.36 | 24.67 | 25.48 | 23.86 | 8.73 | 14.71 | 19.58 | 15.50 | 38.00 | 71.61 | 31.82 | 3.50 | 3.79 | 38.81 | 39.03 | 24.14 |
| SnapKV | 15.74 | 12.59 | 35.87 | 29.77 | 26.43 | 14.17 | 16.17 | 20.35 | 16.47 | 36.50 | 78.04 | 31.84 | 3.50 | 4.57 | 39.03 | 40.15 | 26.32 |
| PyramidKV | 14.84 | 12.10 | 33.63 | 27.73 | 23.95 | 11.77 | 15.27 | 19.79 | 13.99 | 35.50 | 74.17 | 30.50 | 1.50 | 4.71 | 37.90 | 37.99 | 24.71 |
| LAQ | 18.63 | 13.65 | 41.78 | 34.75 | 29.59 | 16.57 | 18.89 | 20.88 | 19.19 | 44.00 | 79.29 | 34.89 | 2.54 | 4.25 | 39.43 | 41.06 | 28.71 |
| LOOKAHEADKV | 17.38 | 14.92 | 41.39 | 35.46 | 29.22 | 17.47 | 20.13 | 20.78 | 21.24 | 51.50 | 80.27 | 36.19 | 3.00 | 4.17 | 34.75 | 37.68 | **29.10** |
| *KV Cache Size = 256* | | | | | | | | | | | | | | | | | |
| StreamingLLM | 14.74 | 12.39 | 25.28 | 26.71 | 23.87 | 8.88 | 16.99 | 19.67 | 17.97 | 44.50 | 74.85 | 35.96 | 3.50 | 3.90 | 40.37 | 42.19 | 25.74 |
| SnapKV | 16.59 | 13.78 | 38.80 | 32.54 | 28.11 | 16.55 | 18.55 | 20.00 | 19.69 | 41.50 | 79.31 | 33.70 | 4.00 | 4.58 | 39.15 | 41.26 | 28.01 |
| PyramidKV | 15.11 | 13.08 | 37.31 | 32.03 | 25.36 | 12.60 | 16.91 | 20.10 | 17.78 | 40.50 | 76.61 | 32.34 | 3.50 | 4.65 | 37.13 | 40.32 | 26.58 |
| LAQ | 18.31 | 14.64 | 41.83 | 35.34 | 29.61 | 17.27 | 20.68 | 21.21 | 21.37 | 52.00 | 79.62 | 36.99 | 4.04 | 4.17 | 40.44 | 43.37 | 30.06 |
| LOOKAHEADKV | 18.23 | 14.70 | 40.25 | 36.52 | 30.45 | 18.50 | 21.75 | 20.91 | 22.46 | 57.50 | 80.09 | 38.05 | 4.00 | 4.50 | 36.19 | 40.73 | **30.30** |
| *KV Cache Size = 512* | | | | | | | | | | | | | | | | | |
| StreamingLLM | 14.62 | 12.86 | 26.37 | 27.03 | 24.19 | 9.96 | 19.02 | 19.61 | 20.99 | 52.50 | 76.92 | 36.50 | 2.54 | 3.64 | 40.76 | 42.83 | 26.90 |
| SnapKV | 17.58 | 14.17 | 40.91 | 34.57 | 29.19 | 16.74 | 20.32 | 20.78 | 22.10 | 52.50 | 80.29 | 34.65 | 3.00 | 4.58 | 38.20 | 42.59 | 29.51 |
| PyramidKV | 16.55 | 13.48 | 39.67 | 32.62 | 28.38 | 15.59 | 18.50 | 20.87 | 20.54 | 48.50 | 79.30 | 34.43 | 4.00 | 4.50 | 38.79 | 41.42 | 28.57 |
| LAQ | 18.45 | 15.26 | 41.93 | 34.75 | 30.44 | 17.63 | 22.39 | 21.45 | 23.11 | 57.50 | 79.04 | 37.81 | 4.00 | 4.17 | 40.76 | 44.64 | 30.83 |
| LOOKAHEADKV | 18.32 | 14.87 | 41.62 | 36.05 | 30.10 | 18.77 | 23.06 | 21.49 | 23.57 | 63.50 | 80.40 | 38.73 | 3.00 | 4.75 | 37.12 | 42.04 | **31.09** |
| *KV Cache Size = 1024* | | | | | | | | | | | | | | | | | |
| StreamingLLM | 15.07 | 13.49 | 29.51 | 28.66 | 25.17 | 11.51 | 20.86 | 19.85 | 23.62 | 57.00 | 79.36 | 37.59 | 4.00 | 3.64 | 39.54 | 44.15 | 28.31 |
| SnapKV | 18.12 | 14.66 | 40.52 | 29.85 | 28.86 | 18.86 | 22.38 | 21.12 | 24.16 | 60.50 | 81.16 | 35.61 | 3.00 | 4.62 | 38.58 | 43.47 | 30.73 |
| PyramidKV | 15.94 | 14.15 | 39.34 | 34.16 | 28.37 | 16.68 | 20.27 | 21.11 | 23.14 | 56.00 | 79.93 | 35.36 | 1.50 | 4.58 | 38.48 | 42.06 | 29.44 |
| LAQ | 19.18 | 15.23 | 41.55 | 34.43 | 30.52 | 18.35 | 23.96 | 21.47 | 24.51 | 61.50 | 79.78 | 38.53 | 3.00 | 4.42 | 40.60 | 45.30 | 31.40 |
| LOOKAHEADKV | 18.51 | 15.41 | 41.49 | 35.41 | 29.74 | 19.29 | 24.93 | 21.19 | 24.58 | 64.00 | 81.07 | 39.06 | 3.50 | 4.80 | 37.99 | 43.55 | **31.53** |
| *KV Cache Size = 2048* | | | | | | | | | | | | | | | | | |
| StreamingLLM | 17.10 | 14.71 | 31.30 | 31.33 | 26.60 | 11.20 | 22.94 | 20.21 | 24.89 | 59.00 | 79.81 | 38.02 | 4.04 | 4.00 | 39.49 | 44.55 | 29.32 |
| SnapKV | 17.73 | 15.74 | 42.03 | 36.12 | 29.48 | 19.34 | 24.30 | 21.75 | 25.22 | 62.50 | 80.90 | 38.27 | 3.00 | 4.75 | 38.28 | 43.52 | 31.43 |
| PyramidKV | 18.83 | 14.50 | 41.40 | 35.75 | 28.89 | 17.40 | 22.05 | 21.14 | 24.97 | 60.50 | 80.62 | 37.33 | 2.50 | 4.50 | 38.51 | 43.46 | 30.77 |
| LAQ | 19.03 | 15.61 | 40.93 | 34.10 | 30.23 | 18.99 | 25.75 | 21.55 | 25.49 | 64.50 | 79.73 | 38.66 | 3.50 | 4.33 | 40.35 | 45.65 | 31.78 |
| LOOKAHEADKV | 18.18 | 16.08 | 42.13 | 35.45 | 30.13 | 19.89 | 26.34 | 21.23 | 25.63 | 64.00 | 80.90 | 39.52 | 3.00 | 4.70 | 38.06 | 44.13 | **31.84** |

Table 10: LongBench evaluation results for Qwen3-1.7B

| | Single-Document QA | | | Multi-Document QA | | | Summarization | | | Few-shot Learning | | | Synthetic | | Code | | Avg. |
|---|---|---|---|---|---|---|---|---|---|---|---|---|---|---|---|---|---|
| | NrtQA | Qasper | MF-en | HotpotQA | 2WikiMQA | Musique | GovReport | QMSum | MultiNews | TREC | TriviaQA | SAMSum | PCount | Pre | Lcc | RB-P | |
| FullKV | 18.94 | 24.78 | 46.15 | 39.15 | 33.11 | 19.03 | 30.33 | 23.06 | 25.18 | 74.00 | 85.21 | 42.87 | 0.00 | 94.50 | 46.82 | 40.24 | 40.21 |
| *KV Cache Size = 64* | | | | | | | | | | | | | | | | | |
| StreamingLLM | 12.25 | 16.83 | 20.38 | 22.62 | 24.61 | 7.26 | 11.43 | 18.96 | 12.12 | 39.00 | 66.48 | 35.14 | 0.00 | 12.50 | 42.95 | 36.90 | 23.71 |
| SnapKV | 13.01 | 17.50 | 30.47 | 28.14 | 25.68 | 8.63 | 11.89 | 19.77 | 11.03 | 37.50 | 78.59 | 37.62 | 0.00 | 54.50 | 44.42 | 36.92 | 28.48 |
| PyramidKV | 13.20 | 16.31 | 27.19 | 23.83 | 25.80 | 6.95 | 11.23 | 19.12 | 10.72 | 37.00 | 73.99 | 35.99 | 0.00 | 43.00 | 43.01 | 35.97 | 26.46 |
| LAQ | 16.64 | 16.99 | 43.66 | 34.19 | 31.41 | 14.00 | 16.09 | 20.94 | 13.76 | 48.00 | 83.63 | 38.89 | 0.00 | 70.75 | 42.51 | 39.08 | 33.16 |
| LOOKAHEADKV | 19.12 | 21.73 | 44.25 | 37.18 | 33.01 | 15.98 | 21.42 | 22.53 | 19.73 | 56.50 | 85.56 | 41.72 | 0.00 | 89.00 | 44.58 | 38.05 | **36.90** |
| *KV Cache Size = 128* | | | | | | | | | | | | | | | | | |
| StreamingLLM | 13.58 | 16.57 | 22.67 | 22.16 | 24.16 | 7.54 | 13.01 | 19.26 | 14.80 | 44.50 | 70.96 | 37.72 | 0.00 | 12.50 | 46.07 | 38.30 | 25.24 |
| SnapKV | 15.94 | 18.73 | 38.02 | 33.35 | 28.02 | 11.74 | 15.40 | 20.80 | 15.71 | 47.00 | 81.65 | 38.77 | 0.00 | 86.00 | 45.47 | 38.55 | 33.45 |
| PyramidKV | 14.13 | 18.14 | 35.49 | 30.97 | 26.99 | 10.53 | 14.27 | 20.50 | 14.29 | 45.50 | 78.72 | 38.06 | 0.00 | 78.00 | 44.05 | 37.95 | 31.72 |
| LAQ | 18.38 | 21.08 | 45.04 | 38.04 | 33.52 | 15.36 | 20.17 | 22.70 | 18.78 | 62.50 | 85.21 | 41.79 | 0.14 | 92.00 | 42.94 | 40.84 | 37.41 |
| LOOKAHEADKV | 19.46 | 23.27 | 44.59 | 37.81 | 33.82 | 17.97 | 23.71 | 23.12 | 21.70 | 65.50 | 85.56 | 42.38 | 0.12 | 92.75 | 44.80 | 38.61 | **38.45** |
| *KV Cache Size = 256* | | | | | | | | | | | | | | | | | |
| StreamingLLM | 13.41 | 17.66 | 22.58 | 23.88 | 23.88 | 7.65 | 16.15 | 19.24 | 17.97 | 47.50 | 76.22 | 40.17 | 0.00 | 13.50 | 46.27 | 37.64 | 26.47 |
| SnapKV | 17.52 | 20.72 | 39.73 | 34.12 | 30.25 | 14.94 | 19.06 | 21.99 | 19.43 | 61.00 | 83.82 | 38.53 | 0.00 | 94.50 | 44.98 | 39.87 | 36.28 |
| PyramidKV | 17.07 | 19.74 | 38.19 | 33.18 | 29.09 | 13.95 | 17.92 | 21.26 | 17.67 | 56.00 | 81.47 | 39.26 | 0.00 | 91.50 | 45.19 | 38.23 | 34.98 |
| LAQ | 18.74 | 21.71 | 45.85 | 37.93 | 33.55 | 16.16 | 22.63 | 23.05 | 21.51 | 70.00 | 85.21 | 42.73 | 0.17 | 95.00 | 41.83 | 41.63 | 38.61 |
| LOOKAHEADKV | 19.60 | 24.30 | 45.69 | 38.81 | 34.02 | 17.91 | 25.51 | 23.11 | 23.15 | 70.00 | 85.37 | 42.16 | 0.12 | 91.00 | 45.29 | 38.98 | **39.06** |
| *KV Cache Size = 512* | | | | | | | | | | | | | | | | | |
| StreamingLLM | 13.92 | 18.40 | 25.10 | 24.91 | 24.77 | 7.67 | 20.38 | 19.63 | 20.78 | 61.00 | 81.80 | 40.38 | 0.00 | 12.50 | 47.28 | 38.62 | 28.57 |
| SnapKV | 19.32 | 22.22 | 44.07 | 36.25 | 30.04 | 15.66 | 22.20 | 22.15 | 21.73 | 70.50 | 84.81 | 40.54 | 0.14 | 94.50 | 46.75 | 40.26 | 38.20 |
| PyramidKV | 17.95 | 20.69 | 41.43 | 36.22 | 29.68 | 14.96 | 20.39 | 21.52 | 20.01 | 66.00 | 84.65 | 40.16 | 0.14 | 93.50 | 45.61 | 38.83 | 36.98 |
| LAQ | 16.99 | 22.67 | 46.97 | 38.10 | 33.66 | 16.38 | 24.58 | 23.39 | 22.99 | 71.50 | 85.37 | 42.25 | 0.17 | 94.00 | 40.85 | 40.30 | 38.71 |
| LOOKAHEADKV | 19.04 | 24.66 | 44.68 | 39.04 | 33.66 | 17.64 | 27.46 | 23.31 | 24.17 | 73.00 | 85.37 | 42.87 | 0.17 | 94.00 | 45.27 | 39.25 | **39.60** |
| *KV Cache Size = 1024* | | | | | | | | | | | | | | | | | |
| StreamingLLM | 15.32 | 18.63 | 26.90 | 27.88 | 26.44 | 8.47 | 23.56 | 20.34 | 23.82 | 65.50 | 84.00 | 41.69 | 0.00 | 18.00 | 46.93 | 39.71 | 30.45 |
| SnapKV | 18.68 | 24.05 | 44.25 | 38.57 | 30.72 | 16.63 | 24.97 | 22.28 | 23.62 | 71.50 | 85.26 | 40.43 | 0.50 | 95.00 | 46.39 | 39.03 | 38.87 |
| PyramidKV | 18.76 | 23.44 | 43.47 | 36.96 | 29.98 | 16.21 | 23.08 | 22.21 | 22.68 | 72.50 | 84.76 | 40.19 | 0.50 | 95.50 | 46.89 | 39.98 | 38.57 |
| LAQ | 16.76 | 22.60 | 45.45 | 38.35 | 33.44 | 17.15 | 26.68 | 23.32 | 24.11 | 73.00 | 84.71 | 43.11 | 0.00 | 95.00 | 39.42 | 41.02 | 39.03 |
| LOOKAHEADKV | 19.20 | 25.15 | 44.48 | 39.09 | 32.85 | 18.39 | 29.18 | 23.24 | 25.14 | 73.00 | 84.84 | 43.19 | 0.17 | 95.00 | 46.24 | 39.86 | **39.94** |
| *KV Cache Size = 2048* | | | | | | | | | | | | | | | | | |
| StreamingLLM | 16.32 | 22.12 | 29.96 | 31.34 | 28.13 | 9.37 | 26.32 | 20.96 | 24.58 | 68.50 | 84.59 | 42.84 | 0.00 | 31.00 | 46.63 | 39.92 | 32.66 |
| SnapKV | 19.44 | 24.89 | 45.18 | 38.97 | 32.65 | 17.32 | 27.89 | 22.60 | 24.81 | 72.50 | 85.21 | 42.37 | 0.17 | 95.00 | 46.50 | 40.43 | 39.75 |
| PyramidKV | 19.52 | 24.01 | 44.53 | 39.48 | 31.94 | 16.37 | 26.16 | 22.68 | 24.64 | 72.00 | 84.84 | 41.79 | 0.50 | 95.00 | 46.57 | 40.12 | 39.39 |
| LAQ | 16.42 | 22.65 | 45.81 | 38.66 | 33.44 | 17.38 | 28.79 | 23.41 | 25.14 | 73.00 | 84.71 | 42.94 | 0.00 | 95.00 | 43.38 | 40.30 | 39.44 |
| LOOKAHEADKV | 19.08 | 25.15 | 45.04 | 38.79 | 33.00 | 17.85 | 29.86 | 23.00 | 25.26 | 73.50 | 85.21 | 43.06 | 0.17 | 94.00 | 46.62 | 40.53 | **40.01** |

Table 11: LongBench evaluation results for Llama3.2-3B

| | Single-Document QA | | | Multi-Document QA | | | Summarization | | | Few-shot Learning | | | Synthetic | | Code | | Avg. |
|---|---|---|---|---|---|---|---|---|---|---|---|---|---|---|---|---|---|
| | NrtQA | Qasper | MF-en | HotpotQA | 2WikiMQA | Musique | GovReport | QMSum | MultiNews | TREC | TriviaQA | SAMSum | PCount | Pre | Lcc | RB-P | |
| FullKV | 25.35 | 40.77 | 50.31 | 53.79 | 40.05 | 25.62 | 33.11 | 24.38 | 25.95 | 72.00 | 88.89 | 43.25 | 4.00 | 97.50 | 54.65 | 56.83 | 46.03 |
| *KV Cache Size = 64* | | | | | | | | | | | | | | | | | |
| StreamingLLM | 19.79 | 20.68 | 24.68 | 42.63 | 34.09 | 16.14 | 16.53 | 20.22 | 14.78 | 38.00 | 75.84 | 33.19 | 4.50 | 10.50 | 46.11 | 43.76 | 28.84 |
| SnapKV | 21.88 | 20.56 | 39.12 | 48.33 | 35.39 | 22.39 | 16.37 | 20.30 | 14.41 | 39.50 | 82.17 | 35.05 | 4.50 | 21.50 | 47.96 | 44.30 | 32.11 |
| PyramidKV | 23.06 | 19.20 | 36.26 | 46.34 | 34.81 | 20.17 | 16.17 | 20.96 | 14.02 | 41.00 | 81.93 | 35.08 | 4.00 | 23.00 | 46.82 | 43.94 | 31.67 |
| LAQ | 23.33 | 28.03 | 48.16 | 54.08 | 37.28 | 23.91 | 18.88 | 21.75 | 16.62 | 47.50 | 86.65 | 37.73 | 4.50 | 74.00 | 47.82 | 46.56 | 38.55 |
| **LOOKAHEADKV** | 25.22 | 33.43 | 49.90 | 53.50 | 39.60 | 24.31 | 22.42 | 22.23 | 20.30 | 64.00 | 89.38 | 38.70 | 4.50 | 77.50 | 50.79 | 49.83 | **41.60** |
| *KV Cache Size = 128* | | | | | | | | | | | | | | | | | |
| StreamingLLM | 19.27 | 20.13 | 25.60 | 43.77 | 33.92 | 16.84 | 17.52 | 19.96 | 17.23 | 42.50 | 79.63 | 36.86 | 4.00 | 10.50 | 50.75 | 47.3 | 30.36 |
| SnapKV | 20.59 | 27.04 | 45.17 | 51.15 | 37.77 | 21.49 | 19.64 | 21.89 | 18.22 | 49.50 | 86.62 | 37.35 | 5.50 | 68.00 | 51.51 | 49.81 | 38.20 |
| PyramidKV | 22.19 | 25.35 | 45.65 | 51.67 | 37.41 | 20.93 | 19.29 | 21.45 | 18.37 | 47.50 | 84.95 | 37.59 | 5.50 | 62.00 | 50.54 | 48.46 | 37.43 |
| LAQ | 25.04 | 35.33 | 51.55 | 53.87 | 39.98 | 23.75 | 21.82 | 23.27 | 20.20 | 61.00 | 89.13 | 39.91 | 5.53 | 89.50 | 53.67 | 53.08 | 42.91 |
| **LOOKAHEADKV** | 25.45 | 36.54 | 51.58 | 53.30 | 40.47 | 24.38 | 24.02 | 23.64 | 21.95 | 68.50 | 89.56 | 41.64 | 4.00 | 91.00 | 53.29 | 53.06 | **43.90** |
| *KV Cache Size = 256* | | | | | | | | | | | | | | | | | |
| StreamingLLM | 19.42 | 22.77 | 28.79 | 43.12 | 32.29 | 16.69 | 19.61 | 19.78 | 19.15 | 50.00 | 84.07 | 39.15 | 4.00 | 11.50 | 52.76 | 49.85 | 32.06 |
| SnapKV | 22.44 | 30.24 | 47.71 | 53.97 | 40.23 | 23.01 | 21.73 | 22.52 | 21.03 | 61.50 | 88.81 | 38.69 | 5.50 | 91.50 | 53.49 | 53.56 | 42.25 |
| PyramidKV | 21.35 | 30.12 | 47.93 | 53.26 | 38.78 | 23.27 | 21.26 | 22.61 | 20.57 | 61.00 | 87.58 | 39.01 | 5.50 | 87.00 | 52.95 | 50.72 | 41.43 |
| LAQ | 24.46 | 38.11 | 50.89 | 53.80 | 39.58 | 24.09 | 23.61 | 23.83 | 22.10 | 70.00 | 89.36 | 41.30 | 4.03 | 95.00 | 55.55 | 55.21 | 44.43 |
| **LOOKAHEADKV** | 25.58 | 37.51 | 50.92 | 53.20 | 40.35 | 25.74 | 25.92 | 24.37 | 23.36 | 69.00 | 89.51 | 42.54 | 4.50 | 94.50 | 53.92 | 55.20 | **44.76** |
| *KV Cache Size = 512* | | | | | | | | | | | | | | | | | |
| StreamingLLM | 19.42 | 23.98 | 29.39 | 43.87 | 32.82 | 16.92 | 21.95 | 20.37 | 22.14 | 59.50 | 85.32 | 41.04 | 4.00 | 12.50 | 55.06 | 51.93 | 33.76 |
| SnapKV | 23.50 | 35.97 | 49.42 | 52.38 | 39.49 | 23.50 | 23.68 | 23.47 | 22.97 | 68.50 | 89.16 | 40.43 | 4.00 | 97.00 | 55.01 | 54.87 | 43.96 |
| PyramidKV | 22.85 | 34.18 | 48.20 | 52.36 | 39.98 | 23.36 | 22.98 | 23.38 | 22.48 | 68.50 | 88.41 | 39.90 | 4.50 | 96.50 | 53.68 | 53.67 | 43.43 |
| LAQ | 24.90 | 39.15 | 50.66 | 53.88 | 39.85 | 26.16 | 25.45 | 23.93 | 23.88 | 71.00 | 88.97 | 42.57 | 4.00 | 96.50 | 55.48 | 55.68 | 45.13 |
| **LOOKAHEADKV** | 24.12 | 38.81 | 50.78 | 53.84 | 40.06 | 25.28 | 27.81 | 24.15 | 24.83 | 69.00 | 89.16 | 42.01 | 4.00 | 97.00 | 55.03 | 56.64 | **45.16** |
| *KV Cache Size = 1024* | | | | | | | | | | | | | | | | | |
| StreamingLLM | 21.33 | 27.49 | 31.93 | 44.78 | 35.00 | 16.63 | 24.32 | 21.01 | 24.45 | 61.50 | 85.48 | 40.93 | 4.00 | 16.50 | 55.09 | 53.40 | 35.24 |
| SnapKV | 24.67 | 38.16 | 51.18 | 54.02 | 40.13 | 25.01 | 25.92 | 23.78 | 24.30 | 69.50 | 88.79 | 40.79 | 4.50 | 98.00 | 55.57 | 56.58 | 45.06 |
| PyramidKV | 24.01 | 37.98 | 51.43 | 53.64 | 39.49 | 24.15 | 25.48 | 23.49 | 24.50 | 68.50 | 88.68 | 40.76 | 4.00 | 98.00 | 55.14 | 55.15 | 44.65 |
| LAQ | 24.55 | 40.78 | 49.15 | 53.65 | 40.01 | 26.68 | 27.69 | 24.07 | 25.25 | 71.50 | 88.89 | 43.54 | 4.00 | 97.00 | 54.93 | 56.68 | 45.54 |
| **LOOKAHEADKV** | 24.28 | 39.51 | 51.35 | 53.64 | 40.36 | 25.07 | 29.61 | 24.17 | 25.41 | 69.50 | 88.99 | 42.90 | 4.00 | 98.00 | 55.59 | 56.72 | **45.57** |
| *KV Cache Size = 2048* | | | | | | | | | | | | | | | | | |
| StreamingLLM | 22.36 | 32.37 | 33.12 | 46.79 | 36.00 | 17.95 | 26.86 | 21.68 | 25.78 | 66.00 | 87.26 | 41.68 | 4.00 | 27.50 | 55.48 | 54.82 | 37.48 |
| SnapKV | 24.54 | 40.50 | 51.36 | 53.80 | 40.60 | 25.24 | 28.41 | 23.92 | 25.62 | 71.50 | 88.86 | 42.39 | 4.00 | 98.00 | 55.07 | 56.97 | 45.67 |
| PyramidKV | 24.05 | 39.80 | 51.31 | 53.88 | 40.03 | 24.50 | 27.88 | 24.27 | 25.65 | 70.00 | 88.80 | 42.11 | 4.00 | 98.00 | 54.84 | 55.59 | 45.29 |
| LAQ | 24.80 | 41.44 | 49.50 | 54.05 | 39.88 | 26.13 | 30.09 | 24.44 | 25.80 | 71.50 | 88.89 | 43.11 | 4.00 | 97.00 | 55.44 | 56.77 | **45.80** |
| **LOOKAHEADKV** | 25.23 | 40.67 | 50.00 | 53.79 | 40.06 | 24.87 | 31.30 | 24.18 | 25.88 | 71.00 | 88.99 | 43.49 | 4.00 | 97.00 | 55.12 | 56.77 | 45.77 |

Table 12: LongBench evaluation results for Qwen3-4B

| | Single-Document QA | | | Multi-Document QA | | | Summarization | | | Few-shot Learning | | | Synthetic | | Code | | Avg. |
|---|---|---|---|---|---|---|---|---|---|---|---|---|---|---|---|---|---|
| | NrtQA | Qasper | MF-en | HotpotQA | 2WikiMQA | Musique | GovReport | QMSum | MultiNews | TREC | TriviaQA | SAMSum | PCount | Pre | Lcc | RB-P | |
| FullKV | 27.45 | 43.30 | 54.45 | 55.63 | 43.43 | 31.61 | 32.24 | 24.61 | 25.00 | 73.00 | 88.76 | 43.65 | 0.75 | 96.50 | 64.29 | 61.39 | 47.88 |
| *KV Cache Size = 64* | | | | | | | | | | | | | | | | | |
| StreamingLLM | 12.46 | 23.96 | 25.93 | 38.56 | 33.40 | 19.47 | 13.74 | 19.71 | 13.04 | 39.50 | 75.48 | 34.33 | 0.50 | 64.50 | 51.42 | 48.46 | 32.15 |
| SnapKV | 15.28 | 25.03 | 31.61 | 40.00 | 34.95 | 18.83 | 12.88 | 19.78 | 12.49 | 40.50 | 75.62 | 33.69 | 1.00 | 69.00 | 51.48 | 47.38 | 33.10 |
| PyramidKV | 15.50 | 24.84 | 34.33 | 40.70 | 35.07 | 19.39 | 13.48 | 19.85 | 13.03 | 41.50 | 76.69 | 33.95 | 1.50 | 73.00 | 52.98 | 47.77 | 33.97 |
| LAQ | 16.55 | 30.74 | 46.21 | 40.58 | 38.10 | 18.35 | 14.96 | 20.74 | 14.48 | 43.50 | 71.25 | 34.40 | 1.50 | 81.25 | 53.45 | 48.45 | 35.91 |
| **LOOKAHEADKV** | 20.49 | 37.99 | 51.37 | 54.71 | 42.30 | 30.90 | 22.10 | 22.08 | 18.71 | 58.50 | 88.85 | 39.71 | 1.00 | 92.00 | 55.89 | 52.33 | **43.11** |
| *KV Cache Size = 128* | | | | | | | | | | | | | | | | | |
| StreamingLLM | 15.69 | 23.56 | 26.02 | 38.03 | 32.38 | 18.86 | 14.79 | 19.67 | 15.20 | 45.50 | 78.32 | 37.32 | 0.50 | 65.50 | 55.83 | 51.76 | 33.68 |
| SnapKV | 19.56 | 29.48 | 43.55 | 49.81 | 37.95 | 25.99 | 16.27 | 21.17 | 16.16 | 49.50 | 86.31 | 38.10 | 1.50 | 95.00 | 58.75 | 53.78 | 40.18 |
| PyramidKV | 15.88 | 27.32 | 38.51 | 48.25 | 36.56 | 20.45 | 14.72 | 20.88 | 14.65 | 46.00 | 76.75 | 35.06 | 1.50 | 89.50 | 55.58 | 49.49 | 36.19 |
| LAQ | 21.44 | 37.82 | 53.26 | 54.98 | 42.75 | 32.08 | 20.44 | 23.69 | 18.86 | 60.50 | 87.55 | 40.48 | 2.50 | 93.00 | 60.85 | 57.87 | 44.25 |
| **LOOKAHEADKV** | 25.17 | 40.13 | 52.28 | 55.10 | 43.47 | 31.38 | 24.83 | 24.46 | 21.57 | 67.00 | 88.85 | 41.37 | 1.00 | 96.50 | 61.00 | 57.77 | **45.74** |
| *KV Cache Size = 256* | | | | | | | | | | | | | | | | | |
| StreamingLLM | 15.66 | 25.74 | 28.99 | 37.34 | 33.47 | 19.02 | 17.65 | 20.12 | 18.02 | 49.50 | 81.97 | 38.87 | 1.00 | 66.50 | 59.45 | 55.05 | 35.46 |
| SnapKV | 24.64 | 35.80 | 47.67 | 54.45 | 40.78 | 29.60 | 19.84 | 22.75 | 19.68 | 60.00 | 87.64 | 39.46 | 1.00 | 96.00 | 62.57 | 58.24 | 43.76 |
| PyramidKV | 18.31 | 31.30 | 44.14 | 51.08 | 36.87 | 25.14 | 18.68 | 22.04 | 17.78 | 56.50 | 85.53 | 38.77 | 1.50 | 95.50 | 59.16 | 53.92 | 41.01 |
| LAQ | 26.88 | 40.94 | 53.82 | 55.76 | 43.22 | 31.53 | 23.34 | 24.03 | 21.57 | 68.50 | 87.72 | 41.61 | 2.00 | 93.50 | 62.60 | 62.03 | 46.19 |
| **LOOKAHEADKV** | 26.25 | 41.08 | 53.03 | 55.21 | 43.28 | 31.99 | 27.13 | 25.09 | 23.46 | 71.50 | 88.76 | 41.89 | 1.00 | 96.50 | 63.42 | 60.09 | **46.85** |
| *KV Cache Size = 512* | | | | | | | | | | | | | | | | | |
| StreamingLLM | 18.02 | 27.64 | 30.11 | 39.03 | 33.32 | 20.70 | 21.47 | 20.39 | 21.96 | 60.50 | 85.45 | 40.27 | 0.50 | 59.50 | 62.54 | 57.33 | 37.42 |
| SnapKV | 25.27 | 39.10 | 51.45 | 54.22 | 42.21 | 32.86 | 23.30 | 23.53 | 22.33 | 70.00 | 88.76 | 40.24 | 1.00 | 96.50 | 64.28 | 60.45 | 45.97 |
| PyramidKV | 21.93 | 34.53 | 49.40 | 53.99 | 40.38 | 30.21 | 21.87 | 22.72 | 20.77 | 67.00 | 88.24 | 40.05 | 1.00 | 96.50 | 61.47 | 57.70 | 44.24 |
| LAQ | 26.50 | 42.56 | 53.88 | 55.24 | 43.25 | 32.14 | 25.92 | 24.46 | 23.42 | 73.00 | 87.72 | 42.94 | 1.50 | 93.50 | 62.99 | 61.47 | 46.91 |
| **LOOKAHEADKV** | 26.86 | 41.97 | 53.10 | 55.59 | 43.97 | 32.09 | 29.57 | 25.35 | 24.61 | 72.00 | 88.76 | 42.85 | 1.50 | 96.50 | 63.83 | 60.96 | **47.47** |
| *KV Cache Size = 1024* | | | | | | | | | | | | | | | | | |
| StreamingLLM | 20.48 | 30.08 | 32.30 | 42.20 | 34.23 | 20.65 | 24.81 | 20.84 | 24.19 | 64.50 | 87.39 | 40.95 | 1.00 | 47.00 | 64.74 | 59.17 | 38.41 |
| SnapKV | 25.91 | 41.72 | 52.26 | 56.50 | 43.15 | 32.08 | 26.69 | 24.53 | 24.02 | 71.50 | 88.76 | 41.77 | 1.00 | 96.50 | 64.28 | 61.91 | 47.05 |
| PyramidKV | 25.81 | 39.40 | 51.89 | 53.26 | 42.26 | 32.08 | 25.11 | 23.72 | 23.61 | 70.00 | 88.76 | 41.10 | 1.00 | 96.50 | 63.93 | 61.88 | 46.27 |
| LAQ | 27.40 | 43.93 | 54.30 | 55.95 | 43.42 | 31.66 | 28.18 | 25.16 | 23.42 | 73.00 | 87.77 | 43.33 | 1.75 | 93.25 | 62.54 | 62.00 | 47.42 |
| **LOOKAHEADKV** | 27.47 | 42.45 | 53.70 | 55.64 | 43.85 | 32.40 | 30.68 | 24.98 | 25.21 | 73.00 | 88.76 | 42.96 | 1.00 | 96.50 | 64.61 | 61.89 | **47.82** |
| *KV Cache Size = 2048* | | | | | | | | | | | | | | | | | |
| StreamingLLM | 20.87 | 34.01 | 36.39 | 44.11 | 37.06 | 21.93 | 28.06 | 21.64 | 25.16 | 67.50 | 88.39 | 41.55 | 0.50 | 52.00 | 63.58 | 60.98 | 40.23 |
| SnapKV | 26.80 | 43.04 | 53.50 | 55.54 | 44.01 | 33.33 | 29.49 | 24.64 | 24.86 | 73.00 | 88.76 | 41.94 | 1.25 | 96.50 | 64.10 | 62.08 | 47.68 |
| PyramidKV | 25.74 | 42.42 | 53.91 | 55.34 | 43.12 | 33.06 | 27.70 | 24.21 | 24.74 | 72.00 | 88.76 | 41.54 | 1.25 | 96.50 | 63.81 | 61.78 | 47.24 |
| LAQ | 27.21 | 43.52 | 53.62 | 55.67 | 43.89 | 31.73 | 30.42 | 24.93 | 25.04 | 73.00 | 87.72 | 43.77 | 1.50 | 93.25 | 63.02 | 61.92 | 47.51 |
| **LOOKAHEADKV** | 27.48 | 42.86 | 53.71 | 55.31 | 43.82 | 32.42 | 31.79 | 24.75 | 25.33 | 73.00 | 88.76 | 43.34 | 1.25 | 96.50 | 64.18 | 62.23 | **47.92** |

Table 13: LongBench evaluation results for Llama3.1-8B

| | Single-Document QA | | | Multi-Document QA | | | Summarization | | | Few-shot Learning | | | Synthetic | | Code | | Avg. |
|---|---|---|---|---|---|---|---|---|---|---|---|---|---|---|---|---|---|
| | NrtQA | Qasper | MF-en | HotpotQA | 2WikiMQA | Musique | GovReport | QMSum | MultiNews | TREC | TriviaQA | SAMSum | PCount | Pre | Lcc | RB-P | |
| FullKV | 31.63 | 46.66 | 56.93 | 58.10 | 48.50 | 31.57 | 34.46 | 25.28 | 26.98 | 72.50 | 91.65 | 43.79 | 6.64 | 99.50 | 65.12 | 58.78 | 49.88 |
| *KV Cache Size = 64* | | | | | | | | | | | | | | | | | |
| StreamingLLM | 25.75 | 21.75 | 31.22 | 49.09 | 42.11 | 23.98 | 17.29 | 20.99 | 16.04 | 38.50 | 82.81 | 34.50 | 7.50 | 99.50 | 54.27 | 48.14 | 38.34 |
| SnapKV | 27.37 | 24.99 | 41.77 | 54.27 | 45.27 | 27.52 | 16.75 | 21.73 | 16.32 | 39.00 | 86.32 | 36.58 | 7.50 | 98.50 | 55.57 | 48.07 | 40.47 |
| PyramidKV | 24.25 | 22.87 | 41.03 | 53.07 | 43.55 | 26.36 | 16.46 | 21.52 | 15.61 | 38.50 | 81.95 | 36.68 | 7.50 | 99.50 | 54.40 | 47.20 | 39.40 |
| LAQ | 27.62 | 33.71 | 52.35 | 55.85 | 48.92 | 28.06 | 19.74 | 23.19 | 18.90 | 46.00 | 88.29 | 40.62 | 6.83 | 100.00 | 55.55 | 51.49 | 43.57 |
| SpecKV | 24.87 | 26.57 | 51.22 | 55.29 | 46.57 | 25.42 | 19.78 | 22.29 | 19.20 | 33.50 | 85.12 | 39.14 | 8.50 | 97.00 | 57.78 | 57.19 | 41.84 |
| **LOOKAHEADKV** | 30.62 | 41.46 | 55.77 | 56.42 | 48.56 | 30.30 | 23.54 | 24.08 | 21.23 | 60.50 | 91.62 | 42.56 | 7.50 | 99.50 | 58.74 | 53.86 | **46.64** |
| *KV Cache Size = 128* | | | | | | | | | | | | | | | | | |
| StreamingLLM | 24.95 | 21.50 | 32.56 | 50.67 | 42.89 | 24.31 | 18.49 | 21.25 | 16.04 | 40.50 | 85.57 | 38.28 | 7.50 | 99.50 | 59.03 | 49.72 | 39.68 |
| SnapKV | 29.13 | 28.06 | 51.23 | 56.79 | 45.30 | 27.81 | 19.99 | 23.03 | 19.73 | 46.00 | 89.72 | 40.44 | 7.50 | 99.50 | 59.50 | 52.19 | 43.50 |
| PyramidKV | 27.70 | 28.86 | 52.00 | 56.76 | 46.11 | 28.13 | 19.86 | 22.81 | 20.03 | 44.50 | 88.41 | 39.73 | 7.50 | 99.50 | 59.84 | 51.96 | 43.36 |
| LAQ | 30.48 | 38.31 | 55.73 | 57.50 | 49.13 | 29.67 | 22.42 | 24.20 | 21.59 | 60.50 | 92.09 | 41.04 | 7.25 | 99.50 | 60.54 | 55.83 | 46.61 |
| SpecKV | 29.22 | 29.12 | 54.05 | 56.54 | 46.30 | 29.90 | 22.65 | 23.18 | 21.25 | 52.00 | 90.02 | 42.14 | 8.83 | 99.50 | 61.11 | 61.38 | 45.45 |
| **LOOKAHEADKV** | 31.32 | 42.85 | 56.78 | 57.04 | 47.44 | 30.82 | 25.18 | 24.33 | 23.09 | 65.50 | 92.24 | 42.96 | 7.50 | 99.50 | 61.75 | 55.29 | **47.72** |
| *KV Cache Size = 256* | | | | | | | | | | | | | | | | | |
| StreamingLLM | 25.96 | 24.08 | 33.73 | 50.56 | 42.61 | 23.49 | 20.86 | 21.60 | 20.64 | 46.00 | 87.50 | 41.09 | 7.50 | 99.50 | 61.19 | 51.53 | 41.12 |
| SnapKV | 27.96 | 34.49 | 55.07 | 57.40 | 46.57 | 29.50 | 22.49 | 23.51 | 22.42 | 54.00 | 91.10 | 40.61 | 7.33 | 99.50 | 62.48 | 55.36 | 45.61 |
| PyramidKV | 28.09 | 36.64 | 55.86 | 57.68 | 46.28 | 29.56 | 22.23 | 23.86 | 22.53 | 56.50 | 91.56 | 41.23 | 7.33 | 99.50 | 62.47 | 53.92 | 45.95 |
| LAQ | 31.03 | 43.97 | 55.93 | 57.78 | 49.42 | 30.42 | 24.48 | 24.60 | 23.29 | 68.00 | 92.20 | 42.61 | 7.08 | 100.00 | 62.70 | 58.09 | 48.23 |
| SpecKV | 28.66 | 36.19 | 57.26 | 58.17 | 48.51 | 30.85 | 24.83 | 24.60 | 23.32 | 61.00 | 91.16 | 42.46 | 8.33 | 99.50 | 64.21 | 63.18 | 47.64 |
| **LOOKAHEADKV** | 31.96 | 44.01 | 56.80 | 57.99 | 47.41 | 31.46 | 27.26 | 24.56 | 24.59 | 69.00 | 92.55 | 42.93 | 7.33 | 100.00 | 62.81 | 57.02 | **48.61** |
| *KV Cache Size = 512* | | | | | | | | | | | | | | | | | |
| StreamingLLM | 27.20 | 26.66 | 34.51 | 50.04 | 42.70 | 23.35 | 23.33 | 21.35 | 23.51 | 57.50 | 87.68 | 41.87 | 7.50 | 97.50 | 62.34 | 53.63 | 42.54 |
| SnapKV | 30.08 | 41.24 | 56.84 | 56.92 | 47.75 | 29.67 | 24.58 | 24.47 | 24.23 | 64.00 | 92.35 | 41.38 | 7.17 | 99.50 | 64.72 | 57.12 | 47.63 |
| PyramidKV | 29.50 | 40.46 | 56.47 | 57.30 | 47.55 | 30.34 | 24.26 | 24.46 | 24.00 | 66.50 | 91.32 | 41.64 | 7.20 | 99.50 | 63.65 | 55.49 | 47.48 |
| LAQ | 31.64 | 45.55 | 55.21 | 57.73 | 49.60 | 30.99 | 26.67 | 24.79 | 24.85 | 71.00 | 92.33 | 43.06 | 6.92 | 100.00 | 62.16 | 58.45 | 48.81 |
| SpecKV | 31.12 | 43.77 | 57.22 | 57.51 | 49.32 | 31.06 | 26.34 | 24.61 | 24.90 | 65.00 | 92.13 | 43.32 | 7.00 | 100.00 | 65.31 | 61.89 | 48.78 |
| **LOOKAHEADKV** | 31.39 | 44.92 | 57.56 | 58.56 | 47.72 | 30.82 | 29.24 | 24.82 | 25.83 | 72.50 | 91.92 | 43.39 | 7.08 | 100.00 | 64.87 | 58.36 | **49.31** |
| *KV Cache Size = 1024* | | | | | | | | | | | | | | | | | |
| StreamingLLM | 27.23 | 30.80 | 36.64 | 50.59 | 43.26 | 23.45 | 25.73 | 21.67 | 25.49 | 63.50 | 88.84 | 42.56 | 7.50 | 93.50 | 63.15 | 55.73 | 43.73 |
| SnapKV | 29.64 | 44.60 | 57.30 | 57.62 | 48.31 | 31.18 | 27.57 | 24.17 | 25.84 | 69.50 | 92.04 | 42.78 | 7.08 | 99.50 | 64.57 | 58.46 | 48.76 |
| PyramidKV | 30.79 | 44.91 | 56.65 | 58.13 | 48.17 | 30.56 | 26.65 | 24.53 | 25.88 | 68.00 | 91.78 | 42.20 | 6.83 | 99.50 | 64.41 | 57.77 | 48.55 |
| LAQ | 31.63 | 45.63 | 55.02 | 57.70 | 50.27 | 31.28 | 28.82 | 25.10 | 24.85 | 72.50 | 92.33 | 43.31 | 6.50 | 100.00 | 62.75 | 59.04 | 49.25 |
| SpecKV | 31.59 | 45.44 | 57.98 | 57.51 | 49.16 | 31.95 | 28.67 | 24.95 | 25.77 | 67.50 | 92.23 | 43.94 | 6.00 | 99.50 | 65.21 | 62.30 | 49.36 |
| **LOOKAHEADKV** | 31.14 | 46.04 | 57.77 | 58.22 | 48.43 | 30.72 | 30.75 | 25.31 | 26.66 | 72.50 | 91.92 | 43.39 | 7.08 | 100.00 | 64.87 | 58.36 | **49.57** |
| *KV Cache Size = 2048* | | | | | | | | | | | | | | | | | |
| StreamingLLM | 28.53 | 37.02 | 39.90 | 51.22 | 45.83 | 23.69 | 28.41 | 21.91 | 26.50 | 67.50 | 90.98 | 42.53 | 7.25 | 90.50 | 64.88 | 57.52 | 45.26 |
| SnapKV | 31.22 | 46.14 | 56.94 | 58.12 | 48.21 | 31.74 | 30.24 | 24.81 | 26.78 | 71.50 | 91.49 | 43.16 | 6.38 | 99.50 | 64.98 | 58.80 | 49.38 |
| PyramidKV | 31.37 | 46.01 | 56.61 | 58.02 | 48.21 | 31.50 | 29.73 | 24.70 | 26.57 | 71.50 | 91.65 | 42.83 | 6.64 | 99.50 | 64.94 | 58.32 | 49.26 |
| LAQ | 31.30 | 45.69 | 55.62 | 57.61 | 49.91 | 31.33 | 30.96 | 25.51 | 26.77 | 72.50 | 92.33 | 43.54 | 6.83 | 100.00 | 63.77 | 59.28 | 49.56 |
| SpecKV | 31.88 | 46.64 | 57.39 | 57.97 | 48.80 | 32.72 | 30.96 | 25.38 | 26.82 | 71.00 | 91.48 | 43.65 | 5.88 | 99.50 | 65.79 | 61.16 | 49.81 |
| **LOOKAHEADKV** | 31.01 | 46.37 | 57.24 | 58.15 | 48.31 | 31.12 | 32.56 | 25.22 | 27.07 | 72.50 | 91.48 | 43.56 | 6.38 | 99.50 | 64.96 | 59.13 | 49.66 |

Table 14: LongBench evaluation results for Qwen3-8B

| | Single-Document QA | | | Multi-Document QA | | | Summarization | | | Few-shot Learning | | | Synthetic | | Code | | Avg. |
|---|---|---|---|---|---|---|---|---|---|---|---|---|---|---|---|---|---|
| | NrtQA | Qasper | MF-en | HotpotQA | 2WikiMQA | Musique | GovReport | QMSum | MultiNews | TREC | TriviaQA | SAMSum | PCount | Pre | Lcc | RB-P | |
| FullKV | 26.04 | 47.76 | 53.33 | 59.23 | 43.37 | 36.05 | 33.66 | 24.05 | 24.79 | 71.50 | 90.21 | 44.43 | 2.00 | 100.00 | 69.39 | 65.57 | 49.46 |
| *KV Cache Size = 64* | | | | | | | | | | | | | | | | | |
| StreamingLLM | 16.62 | 25.37 | 25.56 | 40.57 | 33.95 | 18.48 | 13.97 | 18.98 | 12.40 | 39.50 | 80.65 | 35.10 | 1.50 | 69.00 | 59.25 | 53.13 | 34.00 |
| SnapKV | 15.87 | 28.01 | 35.93 | 41.97 | 33.93 | 21.23 | 14.24 | 19.01 | 12.20 | 42.00 | 80.85 | 32.86 | 3.50 | 69.00 | 58.23 | 51.93 | 35.05 |
| PyramidKV | 15.45 | 27.42 | 36.67 | 42.79 | 34.00 | 19.69 | 14.67 | 18.95 | 12.89 | 43.00 | 80.62 | 33.89 | 2.00 | 71.50 | 58.34 | 52.26 | 35.26 |
| LAQ | 16.22 | 32.11 | 45.02 | 42.35 | 37.53 | 21.07 | 15.71 | 19.56 | 13.47 | 44.00 | 76.64 | 35.15 | 3.50 | 83.50 | 59.06 | 52.04 | 37.31 |
| SpecKV | 15.30 | 27.73 | 44.79 | 37.60 | 36.56 | 12.94 | 16.55 | 19.59 | 14.67 | 32.00 | 58.10 | 32.81 | 4.50 | 74.50 | 60.22 | 54.75 | 33.91 |
| **LOOKAHEADKV** | 22.11 | 43.13 | 51.85 | 59.01 | 42.50 | 34.34 | 22.66 | 21.61 | 18.49 | 64.50 | 88.75 | 39.48 | 1.50 | 67.75 | 62.78 | 56.07 | **43.53** |
| *KV Cache Size = 128* | | | | | | | | | | | | | | | | | |
| StreamingLLM | 17.65 | 26.69 | 28.40 | 41.05 | 33.46 | 20.82 | 15.72 | 19.15 | 15.14 | 43.00 | 82.57 | 38.44 | 1.50 | 70.00 | 62.86 | 56.69 | 35.82 |
| SnapKV | 19.14 | 32.65 | 45.99 | 54.81 | 35.78 | 26.59 | 17.66 | 20.83 | 16.04 | 49.50 | 87.10 | 38.90 | 3.50 | 99.50 | 64.62 | 58.29 | 42.13 |
| PyramidKV | 15.57 | 30.19 | 41.84 | 46.01 | 35.73 | 19.57 | 16.51 | 19.67 | 14.86 | 47.00 | 83.51 | 35.56 | 2.50 | 92.00 | 62.14 | 53.07 | 38.48 |
| LAQ | 22.74 | 42.15 | 53.55 | 57.89 | 42.84 | 36.74 | 21.33 | 22.25 | 18.31 | 64.50 | 89.55 | 40.93 | 3.40 | 100.00 | 66.74 | 61.70 | 46.52 |
| SpecKV | 23.03 | 37.14 | 53.58 | 56.77 | 42.24 | 31.82 | 21.33 | 22.86 | 19.04 | 60.00 | 88.31 | 41.50 | 3.50 | 100.00 | 66.82 | 61.96 | 45.62 |
| **LOOKAHEADKV** | 26.06 | 44.30 | 53.24 | 58.78 | 42.79 | 35.89 | 25.29 | 22.95 | 21.13 | 66.50 | 88.95 | 41.64 | 3.50 | 99.50 | 65.95 | 62.88 | **47.46** |
| *KV Cache Size = 256* | | | | | | | | | | | | | | | | | |
| StreamingLLM | 18.18 | 28.53 | 28.52 | 42.81 | 33.58 | 21.34 | 18.63 | 19.20 | 17.76 | 48.00 | 85.58 | 40.08 | 1.00 | 69.00 | 65.50 | 59.41 | 37.32 |
| SnapKV | 23.03 | 38.32 | 51.04 | 57.36 | 40.67 | 32.82 | 21.51 | 21.89 | 18.97 | 59.50 | 89.46 | 41.06 | 2.00 | 100.00 | 67.62 | 61.88 | 45.45 |
| PyramidKV | 18.47 | 34.87 | 47.44 | 55.68 | 37.89 | 26.67 | 20.43 | 20.92 | 17.43 | 58.50 | 85.20 | 38.98 | 3.50 | 100.00 | 65.51 | 57.32 | 43.05 |
| LAQ | 26.00 | 45.44 | 53.84 | 57.00 | 43.53 | 36.62 | 24.22 | 23.38 | 20.38 | 70.00 | 89.05 | 42.47 | 3.00 | 100.00 | 68.17 | 64.03 | 47.95 |
| SpecKV | 22.58 | 41.09 | 53.89 | 59.85 | 42.34 | 34.50 | 24.53 | 23.64 | 21.25 | 64.00 | 88.13 | 43.12 | 3.00 | 100.00 | 68.39 | 64.40 | 47.42 |
| **LOOKAHEADKV** | 25.88 | 45.40 | 52.68 | 58.47 | 44.05 | 36.13 | 27.77 | 23.71 | 22.88 | 69.00 | 89.05 | 43.32 | 2.00 | 100.00 | 67.83 | 64.71 | **48.31** |
| *KV Cache Size = 512* | | | | | | | | | | | | | | | | | |
| StreamingLLM | 18.94 | 30.86 | 30.21 | 43.89 | 33.26 | 22.51 | 22.24 | 19.62 | 21.16 | 58.50 | 87.48 | 41.11 | 2.00 | 57.00 | 67.06 | 61.59 | 38.59 |
| SnapKV | 24.63 | 43.72 | 51.96 | 57.57 | 42.36 | 34.04 | 25.03 | 22.55 | 21.66 | 69.00 | 89.53 | 42.06 | 3.00 | 100.00 | 69.37 | 64.96 | 47.64 |
| PyramidKV | 23.12 | 40.52 | 51.43 | 57.77 | 40.89 | 32.85 | 23.85 | 21.92 | 19.70 | 68.50 | 89.55 | 40.89 | 3.00 | 100.00 | 67.47 | 61.73 | 46.44 |
| LAQ | 27.34 | 46.98 | 53.70 | 57.31 | 43.35 | 37.64 | 26.93 | 23.67 | 22.19 | 72.50 | 88.96 | 43.81 | 3.00 | 100.00 | 68.23 | 64.71 | 48.77 |
| SpecKV | 24.22 | 45.65 | 54.34 | 60.53 | 43.85 | 35.26 | 27.21 | 24.04 | 22.53 | 70.50 | 90.20 | 43.71 | 3.50 | 100.00 | 69.25 | 65.85 | 48.79 |
| **LOOKAHEADKV** | 25.33 | 46.49 | 52.04 | 59.32 | 43.09 | 36.92 | 29.56 | 23.80 | 24.01 | 71.50 | 90.21 | 44.20 | 2.00 | 100.00 | 68.88 | 65.58 | **48.93** |
| *KV Cache Size = 1024* | | | | | | | | | | | | | | | | | |
| StreamingLLM | 21.25 | 32.82 | 31.44 | 45.94 | 34.38 | 23.34 | 25.73 | 20.25 | 23.50 | 62.00 | 88.71 | 41.18 | 0.50 | 44.00 | 68.39 | 63.65 | 39.19 |
| SnapKV | 24.26 | 46.13 | 52.48 | 58.52 | 42.66 | 36.89 | 28.39 | 23.61 | 23.33 | 69.00 | 89.55 | 43.13 | 2.00 | 100.00 | 69.05 | 66.27 | 48.45 |
| PyramidKV | 23.77 | 42.89 | 53.01 | 58.86 | 42.32 | 35.47 | 27.32 | 23.07 | 22.72 | 71.00 | 89.95 | 42.56 | 2.00 | 100.0 | 68.81 | 64.25 | 48.00 |
| LAQ | 26.11 | 47.27 | 53.45 | 57.01 | 43.52 | 37.26 | 29.33 | 23.88 | 22.47 | 71.50 | 89.63 | 44.00 | 2.00 | 100.00 | 67.94 | 64.83 | 48.84 |
| SpecKV | 24.98 | 46.56 | 54.07 | 59.04 | 43.37 | 34.12 | 29.32 | 24.18 | 23.68 | 71.00 | 90.11 | 44.56 | 3.00 | 100.00 | 69.09 | 66.53 | 48.98 |
| **LOOKAHEADKV** | 25.36 | 47.23 | 52.56 | 59.30 | 43.25 | 36.39 | 31.65 | 23.72 | 24.61 | 71.00 | 90.21 | 44.69 | 0.50 | 100.00 | 68.93 | 65.22 | **49.04** |
| *KV Cache Size = 2048* | | | | | | | | | | | | | | | | | |
| StreamingLLM | 21.73 | 38.54 | 38.02 | 47.96 | 36.78 | 25.53 | 28.69 | 21.47 | 24.11 | 65.00 | 90.30 | 42.85 | 1.00 | 48.50 | 68.37 | 64.63 | 41.47 |
| SnapKV | 25.55 | 47.52 | 53.20 | 58.73 | 42.70 | 36.08 | 30.64 | 23.78 | 24.40 | 71.50 | 90.21 | 43.27 | 1.10 | 100.00 | 69.33 | 65.36 | 48.96 |
| PyramidKV | 25.47 | 46.69 | 53.21 | 58.41 | 42.90 | 36.61 | 29.41 | 23.61 | 24.18 | 71.50 | 90.05 | 42.90 | 1.00 | 100.00 | 69.28 | 65.21 | 48.78 |
| LAQ | 24.94 | 47.22 | 53.71 | 57.72 | 43.45 | 37.40 | 31.24 | 24.03 | 24.51 | 72.50 | 90.13 | 44.54 | 2.00 | 100.00 | 68.29 | 64.68 | 49.15 |
| SpecKV | 24.86 | 47.00 | 53.80 | 61.43 | 43.74 | 34.94 | 31.38 | 23.98 | 24.57 | 70.50 | 91.11 | 44.46 | 0.00 | 100.00 | 69.12 | 65.87 | 49.17 |
| **LOOKAHEADKV** | 26.76 | 48.01 | 52.92 | 59.43 | 43.20 | 36.21 | 32.64 | 23.93 | 24.93 | 71.00 | 90.21 | 44.74 | 1.00 | 100.00 | 69.23 | 65.01 | **49.33** |

## E.5 RESULTS ON RULER

We report the RULER results across all six models tested, with cache budget settings at 64 (Figure 9) and 128 (Figure 10).

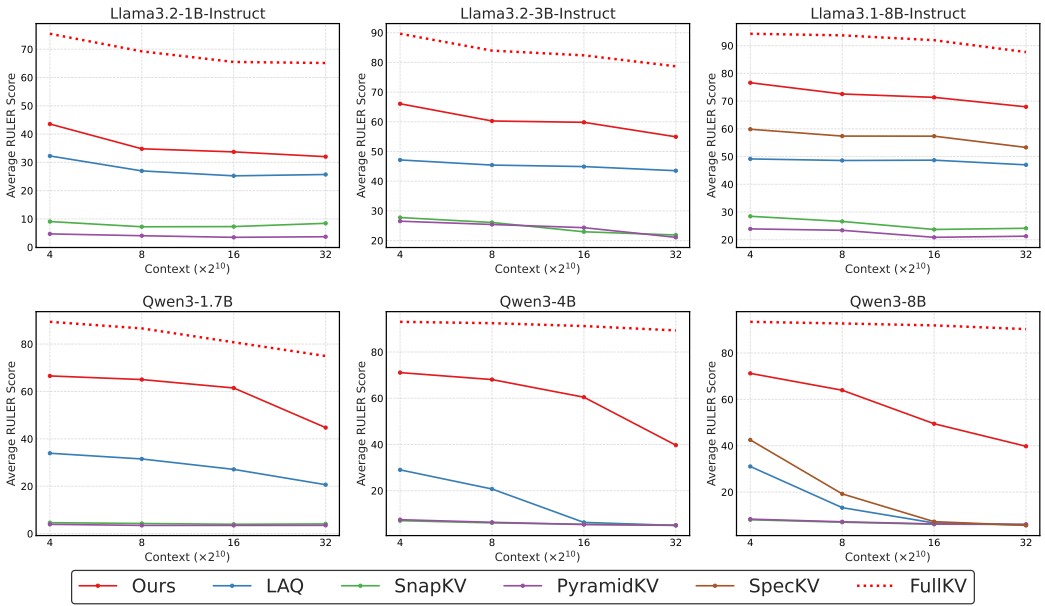

Figure 9: Full RULER results across context lengths (budget = 64)

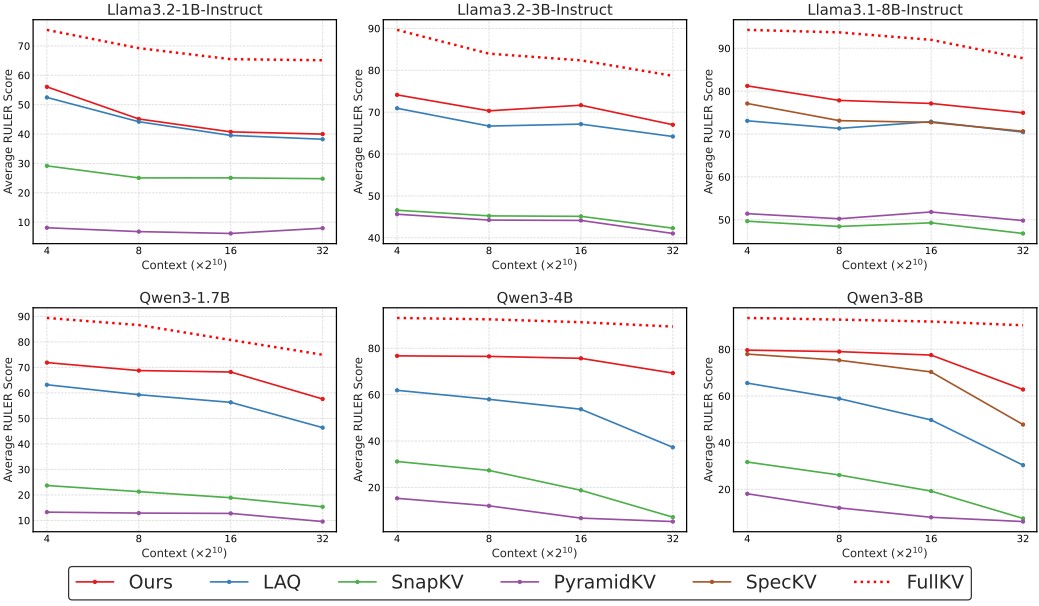

Figure 10: Full RULER results across context lengths (budget = 128)

### E.6 ADDITIONAL EFFICIENCY ANALYSIS

We show the full results of the latency analysis that were omitted in the main paper due to space limitations in this section. Note that the empirical TTFT overheads for some methods can be larger than theoretical estimations. These are probably due to a combination of measurement noise and inefficient implementation of these methods in KVCache-Factory or their official implementations. Better implementations may reduce these overheads significantly, more in line with the theoretical cost.

Table 15: Theoretical and Practical Analysis across various context lengths and methods.

| Context Length | Method | Theoretical Cost | | | | Empirical Cost | |
| | | Compute (TFLOPs) | Memory Traffic (GB) | TTFT (ms) | TTFT Overhead (ms) | TTFT (ms) | TTFT Overhead (ms) |
| --- | --- | --- | --- | --- | --- | --- | --- |
| 4K | Forward Pass Only | 60 | 13 | 113 | N/A | 130 | N/A |
| | LOOKAHEADKV | 60 | 13 | 114 | 0.92 | 141 | 11.38 |
| | SnapKV | 60 | 13 | 113 | 0.01 | 143 | 13.14 |
| | SpecKV | 70 | 77 | 165 | 52.10 | 223 | 92.42 |
| | LAQ | 61 | 444 | 347 | 233.81 | 637 | 506.58 |
| 8K | Forward Pass Only | 136 | 13 | 257 | N/A | 291 | N/A |
| | LOOKAHEADKV | 137 | 13 | 258 | 1.03 | 302 | 10.88 |
| | SnapKV | 136 | 13 | 257 | 0.01 | 311 | 20.17 |
| | SpecKV | 159 | 81 | 337 | 79.53 | 411 | 120.51 |
| | LAQ | 137 | 445 | 492 | 234.59 | 800 | 509.38 |
| 16K | Forward Pass Only | 336 | 13 | 635 | N/A | 658 | N/A |
| | LOOKAHEADKV | 337 | 13 | 636 | 1.27 | 677 | 18.50 |
| | SnapKV | 336 | 13 | 635 | 0.01 | 695 | 37.12 |
| | SpecKV | 398 | 89 | 792 | 157.05 | 866 | 207.31 |
| | LAQ | 337 | 447 | 871 | 236.15 | 1182 | 523.54 |
| 32K | Forward Pass Only | 928 | 13 | 1754 | N/A | 1760 | N/A |
| | LOOKAHEADKV | 929 | 13 | 1755 | 1.74 | 1798 | 38.04 |
| | SnapKV | 928 | 13 | 1754 | 0.01 | 1838 | 77.67 |
| | SpecKV | 1115 | 106 | 2156 | 402.80 | 2263 | 502.87 |
| | LAQ | 930 | 451 | 1993 | 239.26 | 2314 | 553.68 |

## F  HYPER-PARAMETERS

**Training hyper-parameters.**    Learning rate was searched for Llama and Qwen model family among $[5 \times 10^{-5}, 1 \times 10^{-4}, 2 \times 10^{-4}, 1 \times 10^{-3}]$. The final hyper-parameters for all experiments are shown in Table 16.

Table 16: Training hyperparameters.

| Parameters | Values |
|---|---|
| Optimizer | Adam |
| $\beta_1, \beta_2$ | $0.9, 0.95$ |
| Effective Batch Size | 32 |
| Drop-out ($p$) | 0.0 |
| Max Sequence Length | 16384 (prompt length) + 512 (response length) |
| Train Iters | 7600 |
| Learning rate | $1 \times 10^{-3}$ (for Llama), $2 \times 10^{-4}$ (for Qwen) |
| Schedule | Cosine |
| Warmup steps | 2% |
| Min LR | 0.0 |
| Gradient clipping | 1.0 |

**Eviction hyper-parameters.**  We use the implementations in KVCache-Factory or their official implementations (SpecKV) for all baseline methods, except for LAQ which we re-implement ourselves due to the lack of an official release. Following prior works (Li et al., 2024; Cai et al., 2024; Galim et al., 2026), we use standard configuration settings for all baseline methods, including an observation window size of 32, maxpooling kernel size of 7, and mean reduction for GQA compatibility (Feng et al., 2024). For LOOKAHEADKV, we use the same settings, except we do not use window size, as our method does not train with the suffix window for prediction. Further, since our lookahead size $n_{\text{lookahead}}$ is 32, we set the maximum generation limit of LAQ and SpecKV to 32 tokens so that the methods can be compared using the same number of draft tokens.

## G  DATASETS, BENCHMARKS, AND SOFTWARE

**Software** Our source code is available in the supplementary, and our implementation is built on KVCache-Factory.

**Training Dataset** Our training dataset mixture consist of random samples from publicly available datasets: 50K long_sft subset of ChatQA2-Long-SFT-data, 20K subset of tulu-3-sft-olmo-2-mixture, 7K samples from The Stack, and 3K samples from MetaMathFewshot, HellaSwag_DPO_Fewshot, and ARC_DPO_Fewshot, respectively.

**Evaluation Benchmarks** We used LongBench dataset as fetched and processed by KVCache-Factory, see HF Dataset for the official source. For RULER, we used RULER Github. For LongProc, we used LongProc Github.

## H  LLM USAGE

LLM assistants were used to refine the wording of selected sentences, while the majority of the text was written by human. All LLM-generated text was carefully inspected to ensure that it contained no harmful or controversial content. Additionally, we used LLMs to help in finding some of the related literature discussed in the paper.

