# OpenReview forum: "LookaheadKV: Fast and Accurate KV Cache Eviction by Glimpsing into the Future without Generation"
_ICLR.cc/2026/Conference — ICLR 2026 Poster_

### Official Review · Reviewer_4uAb · 2025-10-27

**Soundness:** 4
**Presentation:** 4
**Contribution:** 3
**Rating:** 8
**Confidence:** 4

**Summary:**

This paper proposes to use LoRA and learnable special tokens during the attention step. These components assist the prefill phase in determining which parts of the KV cache should be retained and which can be evicted. The approach ensures accuracy and model performance without significantly increasing decode overhead.

**Strengths:**

1. The problem is well defined. The paper addresses a clearly defined and critical problem: how to efficiently predict the attention patterns of future tokens in the input context for KV Cache eviction.
2. Concise Solution. The proposed solution is concise and seems to work well according to experiments.
3. Comprehensive Experiments. The paper is supported by a thorough experimental evaluation. The accuracy tests cover multiple models and diverse scenarios, including long-input and long-output tasks. The efficiency analysis provides both theoretical cost models and empirical TTFT measurements.

**Weaknesses:**

1. Limited Practicality for Long-Form Generation: While LKV outperforms baselines on long-output tasks, its absolute performance still degrades significantly (as shown in Section 4.3, "Long-Form output Evaluation").
2. Prefill-Only Eviction Limits Applicability: The method's focus exclusively on prefill-stage eviction is a major limitation. Many SOTA reasoning models rely on a large number of decoding steps, during which the KV cache continues to grow. Ignoring decoding phase significantly reduces its overall utility in these increasingly important use cases.
3. Performance Advantage is Concentrated in Extreme Low-Budget Scenarios: The most significant performance gains of LKV over strong baselines like LAQ are demonstrated in extremely low-budget settings (e.g., a cache budget of 64 or 128). These settings, while useful for stress-testing, are not always representative of practical deployments where budgets might be more generous. In higher-budget scenarios, the performance gap narrows considerably, questioning the method's marginal benefit when resources are less constrained.

**Questions:**

1. Could the authors elaborate on the detailed training cost for proposed LKV? Although this process is a one-time effort, it currently appears that training is required for each different model.
2. The training process utilizes greedy decoding to generate ground-truth. But in practice, the most popular sampling strategy is not always greedy decoding. How does LKV's performance generalize to stochastic decoding strategies such as beam-search or top-p? Is there a risk that the learned modules overfit to the deterministic attention patterns of greedy decoding, leading to a performance drop when the generation path becomes less predictable?

---

> ### Author Response · Authors · 2025-11-21
> **Official Comment by Authors**
>
> Dear reviewer, We sincerely appreciate the time and effort you invested in reviewing our work.
>
> We sincerely appreciate your positive review and acknowledgment that **our paper addresses a clearly defined and critical problem**, our solution is **concise**, and that our experiments are **comprehensive and thorough**, including multiple models and scenarios, and with theoretical and empirical efficiency analyses. Our responses to your comments are provided below.
>
> ### W1) Limited Practicality for Long-Form Generation
> Due to limited memory and compute, we had trained LookaheadKV only with the maximum generation length of 512 tokens, as mentioned in section 4.1. We conjecture that this is the cause of more degradation compared to the FullKV performance when increasing the output sequence length to 2K in the long-form output evaluation task in section 4.3. However, our method outperforms other baselines even at longer sequence lengths, demonstrating LookaheadKV's ability to generalize to longer unseen output sequence lengths.
>
> ### W2) Prefill-Only Eviction Limits Applicability
> Reasoning models typically require much higher compute and generation steps. Furthermore, typical on-device applications usually have long input prompt with relatively shorter response. Due to limited compute, while we only explore applying LookaheadKV for prefill eviction, this method is equally applicable to evicting decoded tokens, even for very long generations. For example, one approach could be to recalculate the importance scores and re-evict unimportant KV cache after every N=512 steps of generation.
>
> ### W3) Performance Advantage is Concentrated in Extreme Low-Budget Scenarios
> Larger KV cache budgets negatively impact generation throughput, as KV cache compression significantly reduces the size of KV cache that must be fetched for each decoded token. For example, in an on-device setting, with the commonly used W4A16 quantization, a cache budget of 128 yields 6% higher tokens/sec than 1024 for Llama-1B, and 23% for the smaller MobileLLM-350M. Smaller models are often used as speculative decoders to accelerate on-device generation, making our method particularly attractive
>
> For batched generation, these advantages can be significantly larger, even for larger models, as the time to fetch model weights is amortized over the batch size. With a batch size of 32 and no quantization, a cache budget of 128 yields 45% higher tokens/sec than 1024 for Llama-1B, an 24% for Llama-8B. Reducing KV cache memory will also enable faster model serving by increasing batching.
>
> In summary, our approach **enables using a very low cache budget** for increased generation throughput, while **maintaining similar performance** to much higher cache budgets of prior methods.
>
> ### Q1-1) Could the authors elaborate on the detailed training cost for proposed LKV?
> In our experimental settings, using one H100 GPU node, applying LookaheadKV training to Llama-1/3/8B models took approximately 5/10/18 hours, respectively. We believe that a more optimized implementation could further reduce the training cost of LookaheadKV.
>
> ### Q1-2) Although this process is a one-time effort, it currently appears that training is required for each different model.
> Yes, it is correct that training is required for each different model. However, we emphasize that the substantial efficiency gains and performance improvements offered by LookaheadKV **far outweighs** the additional training cost, making it a highly advantageous advancement over prior methods. Specifically:
>
> 1. **One‑time training cost**: the training cost for LookaheadKV is incurred only once, after which the cost can be **amortized across all subsequent inference calls**. This contrasts with other draft-based methods that require repeated draft generation for importance estimation, resulting in higher overall expense.
> 2. **Reduced TTFT**: reducing **TTFT is essential for improved user experience**, and is particularly **practical in latency‑sensitive contexts** such as mobile applications.
> 3. **Better performance**: across a broad range of benchmarks, LookaheadKV **consistently outperforms** previous methods, leading to markedly less information loss caused by eviction and delivering more reliable results.
>
> In summary, the overall improvement delivered by LookaheadKV significantly outweights the additional cost of training LookaheadKV.
>
> ### Q2) The training process utilizes greedy decoding to generate ground-truth. But in practice, [...]
> We kindly refer the reviewer to our common response, where we provide a detailed discussion of LookaheadKV’s effectiveness during both training and inference under stochastic decoding settings. We thank the reviewer for raising this point.
>
> -------------------------------------------------------------------------------------------------------------
> We again thank the reviewer for their insightful comments positive feedback, and we hope that our response helps resolve the concerns.

---

> > ### Author Response · Authors · 2025-11-27
> > **Thank you for your feedback**
> >
> > Dear reviewer,
> > thank you again for your insightful and thoughtful comments. We have made our best efforts to give thorough explanations to your comments and updated the manuscripts accordingly.
> >
> > As we are now midway through the discussion period, we just wanted to politely follow up to see if our response has addressed your concerns. We are eager to engage further and would be very grateful for any additional feedback you may have. We sincerely hope you could look through our response and have a further comment at your convenience if you have any questions about the paper.
> >
> > Best wishes,
> > Authors.

---

### Official Review · Reviewer_nWM2 · 2025-10-28

**Soundness:** 3
**Presentation:** 2
**Contribution:** 2
**Rating:** 4
**Confidence:** 4

**Summary:**

The paper proposes LookaheadKV, a training-time method that learns to predict token importance for KV cache eviction without running a draft generation. During prefilling, the model appends a small number of learnable lookahead tokens and activates a lightweight LoRA only on these tokens. The attention induced by these lookahead tokens is trained to match the attention that would be observed if the future response were known. At inference time, the method uses the learned mechanism to rank prompt tokens and evict low-importance items from the KV cache. Experiments on long-context benchmarks report accuracy that is competitive or better than recent baselines, with low overhead at prefilling and no added cost during decoding.

**Strengths:**

The central idea is interesting: use a pretrained module with a small LoRA applied on a short learnable window to substitute for running an explicit generate step. This reduces prefilling cost while still capturing signals about future attention. The method is simple to integrate, has low inference overhead, and shows consistent gains in low-budget regimes. The empirical results cover several model families and tasks, and the latency accounting is practical.

**Weaknesses:**

The presentation has gaps. I can understand the LoRA training and the alignment objective, but I do not understand precisely how the lookahead embeddings are obtained. It is not fully clear whether these lookahead tokens are new learned embeddings, adapted from existing vocabulary embeddings, or derived from another module. The paper would benefit from a precise definition of the parameterization, initialization, and update path of these embeddings.

**Questions:**

1. How long does training take in time for the main models, and on what hardware budget. Do you believe the small TTFT improvements and decoding-time stability justify this training cost. How do you view this trade off in practical deployments.
2. Have you tried combining the importance score from LookaheadKV with a SnapKV score. I am thinking of using both the lookahead tokens and the last few prompt tokens as the observation window, so that the score contains both prompt-side and generate-side information. Do you expect this to help, and can you share preliminary results if any.
3. For extreme extrapolation, for example the 128k context window of Llama 3.2, can you report RULER results at 64k and 128k. Even a reduced subset would be helpful to show the scaling trend.

---

> ### Author Response · Authors · 2025-11-21
> **Official Comment by Authors (1/2)**
>
> Dear reviewer, thank you for your time and effort for reviewing our work.
>
> We sincerely appreciate your acknowledgment that the central idea of our **approach is interesting**, the method is **simple to integrate**, it achieves **lower inference overhead** with **consistent performance gains**, and our results demonstrate **broad coverage across models**, tasks, and **practical latency analyses**. Our responses to your comments are provided below.
>
>
> ### W1) The presentation has gaps. [...]
> We agree that we should have made the description of the lookahead tokens clearer, we provide a more detailed definition below, which we will update in the main paper.
>
> As mentioned in section 4.1, for given input prompt tokens $X$, LookaheadKV appends a sequence of trainable “soft” lookahead tokens $L = [l_1, ..., l_{n_{\text{lookahead}}}]$. These tokens are new tokens added to the vocabulary. Just like all other vocabulary embeddings, the embeddings for these new token ids are of shape $\mathbb{R}^{d_{\text{embed}}}$. These are randomly initialized with default PyTorch initialization.
>
> During the prefill forward pass, these tokens are encoded by the model just like any other token. Since these tokens are appended at the end of the input prompt, the prompt tokens do not attend to these tokens due to causal attention, ensuring that the encodings of the prompt tokens unaltered by the addition of the lookahead tokens.
>
> The average attention scores between the lookahead tokens and the prompt tokens are used as the importance scores for KV cache eviction; hence we would like to minimize the divergence between these importance scores and those obtained from the ground-truth, across all layers and heads. The aggregated loss is backpropagated to update lookahead LoRA and lookahead tokens while all original model parameters are frozen.
>
> During inference, we append the learned lookahead tokens to the input prompt and calculate the importance scores to perform KV cache eviction. These tokens are then removed during decoding, ensuring that subsequent generation is unaffected by the LookaheadKV modules.
>
>
> ### Q1-1) How long does training take in time for the main models, and on what hardware budget?
> In our experimental settings, using one H100 GPU node, applying LookaheadKV training to Llama-1/3/8B models took approximately 5/10/18 hours, respectively. We believe that a more optimized implementation could further reduce the training cost of LookaheadKV.
>
> ### Q1-2) Do you believe the small TTFT improvements and decoding-time stability justify this training cost? [...]
> We emphasize that the substantial efficiency gains and performance improvements offered by LookaheadKV **far outweighs** the additional training cost, making it a highly advantageous advancement over prior methods. Specifically:
>
> 1. **One‑time training cost**: the training cost for LookaheadKV is incurred only once, after which the cost can be **amortized across all subsequent inference calls**. This contrasts with other draft-based methods that require repeated draft generation for importance estimation, resulting in higher overall expense.
> 2. **Reduced TTFT**: reducing **TTFT is essential for improved user experience**, and is particularly **practical in latency‑sensitive contexts** such as mobile applications. We would like to highlight that the TTFT improvement brought by LookaheadKV is in fact **highly significant** rather than "small"; as shown in Table 5 in section E.3 of appendix, using Llama-8B-Instruct at 16K context length for instance, the TTFT of LAQ (1182 ms) is nearly double the TTFT of LookaheadKV (677 ms). This overhead not only noticeably affect user experience, but also results in significantly larger resource usage.
> 3. **Better performance**: across a broad range of benchmarks, LookaheadKV **consistently outperforms** previous methods, leading to markedly less information loss caused by eviction and delivering more reliable results.
>
> In summary, LookaheadKV’s one-time training cost, significant reduction in TTFT, and overall superior performance make it a **highly compelling approach** over previous methods.

---

> > ### Author Response · Authors · 2025-11-21
> > **Official Comment by Authors (2/2)**
> >
> > ### Q2) Have you tried combining the importance score from LookaheadKV with SnapKV score?
> > Following your comment, we evaluated LookaheadKV by also including the last 32 prompt tokens for importance score estimation, as done in SnapKV. As shown below, we observe a slight drop in performance when SnapKV importance scores are included. Unlike prior works, performance degradation when averaging lookaheadKV with SnapKV importance scores, compared to using LookaheadKV scores alone, clearly demonstrates that the importance scores predicted by **our method is superior** to SnapKV.
> >
> >
> > | Method                            | narrativeqa | qasper | multifieldqa_en | hotpotqa | 2wikimqa | musique | gov_report | qmsum | multi_news | trec | triviaqa | samsum | passage_count | passage_retrieval_en | lcc  | repobench‑p | Avg. |
> > |-----------------------------------|------------|--------|-----------------|----------|----------|---------|------------|-------|------------|------|----------|--------|----------------|----------------------|------|-------------|------|
> > | LookaheadKV + SnapKV window       | 17.33      | 13.33  | 40.91           | 35.49    | 29.07    | 16.47   | 17.62      | 20.4  | 18.14      | 48.5 | 80.51    | 35.33  | 2.5            | 4.57                 | 36.68| 39.52       | 28.52 |
> > | LookaheadKV                       | 17.38      | 14.92  | 41.39           | 35.46    | 29.22    | 17.47   | 20.13      | 20.78 | 21.24      | 51.5 | 80.27    | 36.19  | 3              | 4.17                 | 34.75| 37.68       | **29.1**  |
> >
> > *LongBench evaluation results using LookaheadKV only and LookaheadKV + SnapKV window, on Llama3.2-1B-Instruct. Cache budget=64*
> >
> > ### Q3) For extreme extrapolation, for example the 128k context window of Llama 3.2, can you report RULER results at 64k and 128k? [...]
> > Following your comment, we report the RULER evaluation at longer contexts and report the results below. As shown, LookaheadKV achieves the **best performance** at these context lengths, showing that the effectiveness of our method scales to longer context lengths.
> >
> > | RULER | FullKV | LookaheadKV | SnapKV | SpecKV | LAQ |
> > |-------|--------|-------------|--------|--------|-----|
> > | 64k   | 88.48  | **71**          | 36.15  | 65.08  | 64.73 |
> > | 128k  | 77.98  | **55.83**       | 27.64  | 53.16  | 50.91 |
> >
> > *RULER evaluation results at longer context lengths using Llama3.1-8B-Instruct. Cache budget=128*
> >
> > -------------------------------------------------------------------------------------------------------------
> > We thank the reviewer for their insightful comments. We hope that our response helps resolve the concerns, and we respectfully ask the reviewer to reconsider their assessment if they find our clarifications satisfactory.

---

> > > ### Author Response · Authors · 2025-11-27
> > > **Thank you for your feedback**
> > >
> > > Dear reviewer,
> > > thank you again for your insightful and thoughtful comments. We have made our best efforts to give thorough explanations to your comments and updated the manuscripts accordingly.
> > >
> > > As we are now midway through the discussion period, we just wanted to politely follow up to see if our response has addressed your concerns. We are eager to engage further and would be very grateful for any additional feedback you may have. We sincerely hope you could look through our response and have a further comment at your convenience if you have any questions about the paper.
> > >
> > > Best wishes,
> > > Authors.

---

### Official Review · Reviewer_CrG9 · 2025-10-28

**Soundness:** 2
**Presentation:** 3
**Contribution:** 2
**Rating:** 4
**Confidence:** 5

**Summary:**

LookaheadKV proposes a lightweight, training-based framework for predicting token importance during prefill to enable accurate KV-cache eviction without draft generation or additional inference steps. The key idea is to introduce a small set of learnable “lookahead tokens”, activated only during prefill, whose attention interactions approximate the model’s future decoding behavior. These tokens are enhanced by LoRA-based low-rank adapters that learn to predict future attention distributions via a KL-divergence ranking objective, requiring no model modification or retraining of base weights. LookaheadKV achieves near-oracle token importance estimation, outperforming SnapKV, SpecKV, and StreamingLLM across benchmarks.

**Strengths:**

1. Strong empirical results with broad coverage: Outperforms baselines on multiple benchmarks
2. Comprehensive ablations and analysis: Includes detailed experiments on number of lookahead tokens, LoRA layer coverage, training context length, and budget scaling

**Weaknesses:**

1. Unclear generalization to stochastic decoding: The method is trained and evaluated primarily under greedy decoding, assuming deterministic next-token prediction. It remains untested under sampling-based decoding
2. Absence of multi-turn or instruction-following benchmarks: Evaluations mostly use single-turn or synthetic long-context datasets. Multi-turn reasoning or conversational tasks, where future tokens are highly context-dependent, are missing
3. Lack of Subtask-Level Analysis on LongBench: The paper evaluates on LongBench but reports only averaged or composite scores, omitting per-subtask breakdowns

**Questions:**

1. Would LookaheadKV’s importance predictions remain accurate when decoding randomness changes token trajectories?
2. How to select retained tokens with lookahead tokens.
3. What is the breakdown scores for LongBench? How Lookahead performs on different tasks.
4. Do different LLMs need different LoRA?
5. How does this method differ from works such as “Learning to Compress Prompts with Gist Tokens”, which train models to summarize and compress prompt tokens to reduce KV-cache size, rather than to learn predictive lookahead behavior for future token importance?

---

> ### Author Response · Authors · 2025-11-21
> **Official Comment by Authors (1/2)**
>
> Dear reviewer,
> We sincerely appreciate the time and effort you invested in reviewing our work.
>
> We are grateful that you acknowledged that our method demonstrates **strong empirical results** with **broad coverage**, and that **our ablations and analyses are comprehensive**. Our responses to your comments are detailed below.
>
> ### W1) Unclear generalization to stochastic decoding
> We kindly refer the reviewer to our common response, where we provide a detailed discussion of LookaheadKV’s effectiveness during both training and inference under stochastic decoding settings. We thank the reviewer for raising this point.
>
> ### W2) Absence of multi-turn or instruction-following benchmarks
> Following your comment, we evaluate LookaheadKV against other baselines on MT-Bench, a benchmark that evaluates LLM's multi-turn conversation ability, covering a broad range of domains including writing, reasoning, math, etc. We evaluate the Qwen3 model families with various cache budget settings, and use Qwen3-235B-A22B as the judge for evaluation.
>
> Again, across all budgets, **LookaheadKV is either comparable or superior**, especially at lower budgets, compared to all other baselines. This further demonstrates the effectiveness of the method in multi-turn conversational tasks where future tokens are highly context-dependent.
>
> **Qwen‑1.7B, FullKV score: 7.19**
> |  Budget | PyramidKV | SnapKV | StreamingLLM | LookaheadKV | LAQ |
> |---|-----------|--------|--------------|--------|------|
> | 64 | 5.81| 5.95| 5.83| **6.7**| 6.19 |
> |128| 6.38| 6.65| 6.16| **7.12**| 6.91 |
> |256| 6.9| 6.94| 6.91| **7.2**| 7.02 |
> |512| 7.09| 7.03| 7.08| **7.29**| 7.2  |
> |1024| 7.28| 7.13| 7.03| **7.35**| 7.32 |
>
> **Qwen‑4B, FullKV score: 8.02**
> | Budget | PyramidKV | SnapKV | StreamingLLM | LookaheadKV | LAQ | SpecKV |
> |---|-----------|--------|--------------|--------|------|----------|
> | 64| 6.85| 6.6| 6.24| **7.69**| 7.06 | 7.05|
> |128| 7.55| 7.71| 7.24| **8.12**| 7.7  | 7.78|
> |256| 7.9| 8.2| 7.87| 8.06| **8.12** | 8.11|
> |512| **8.15**| 8.12| 8| 8.08| 8.06 | 8.02|
> |1024| 7.9| 8.01| 8.01| **8.02**| 7.89 | 8.09|
>
> **Qwen‑8B, FullKV score: 8.48**
> | Budget | PyramidKV | SnapKV | StreamingLLM | LookaheadKV | LAQ | SpecKV |
> |---|-----------|--------|--------------|--------|------|----------|
> | 64| 7.33| 7.26| 6.81| **8.04**| 7.58 | 7.69|
> |128| 7.85| 7.94| 7.64| **8.41**| 8.24 | 7.97|
> |256| 8.42| 8.43| 8.34| 8.51| **8.56** | 8.45|
> |512| 8.43| 8.36| 8.44| 8.53| **8.63** | 8.5|
> |1024| **8.61**| 8.42| 8.44| 8.48| 8.54 | 8.38|
>
> *MT-Bench performance across Qwen3 models, over multiple cache budgets.*
>
> ### W3) Lack of Subtask-Level Analysis on LongBench
> While Table 4 in section E of appendix had per-subtask scores for Llama3.1-8B-Instruct and Qwen3-8B, we had omitted the full scores for all models for all KV cache budgets for brevity. We have updated the paper with the entire set of scores in appendix section E.
>
> ### Q1) Would LookaheadKV’s importance predictions remain accurate when decoding randomness changes token trajectories?
> As in **W1**, We kindly refer the reviewer to our common response, where we provide a detailed discussion of LookaheadKV’s effectiveness during both training and inference under stochastic decoding settings.
>
> ### Q2) How to select retained tokens with lookahead tokens?
> We should have made the KV selection mechanism ofLookaheadKV clearer. We provide a more detailed definition below, which we will update in the main paper.
>
> We select **Top-K KV pairs according to their estimated importance scores**, which is identical to prior methods (SnapKV, SpecKV, LAQ). The importance scores are estimated for each layer and head by first computing the attention between the queries of lookahead tokens and keys of the input prompt, and then aggregating the matrix to obtain the importance score vector.
>
> ### Q3) What is the breakdown scores for LongBench? How Lookahead performs on different tasks.
> We have updated the paper with the entire set of scores in appendix section E.
>
> ### Q4) Do different LLMs need different LoRA?
> We kindly refer the reviewer to section 4.1, where we describe that we use the same LoRA configurations with rank=8 and alpha=32, applied to all projection and feed-forward modules for all LLMs.

---

> ### Author Response · Authors · 2025-11-21
> **Official Comment by Authors (2/2)**
>
> ### Q5) How does this method differ from works such as “Learning to Compress Prompts with Gist Tokens”?
> Regarding comparison to paper "Learning to Compress Prompts with Gist Tokens"[1], explores "prompt compression" whereas our work explores "KV-Cache Eviction". While similar, there are several key differences -
> 1. [1] requires finetuning the entire model to achieve good generation scores, as the model weights have to completely adapt to generate from the gist tokens. Our method is significantly more lightweight, as all the model weights are frozen -- only the choice of which tokens are attended is changed. Modern LLMs often undergo extensive multi-round post-training, and retraining the model while maintaining original accuracy may be difficult.
> 2. [1] requires compressing all information from the prompt into the gist token embeddings, whereas KV-cache Eviction methods select which tokens to attend to. As shown in [1], Instructions containing crucial specific details may be lost in this process. By keeping the original token kv states, specific knowledge from unevicted input tokens remains fully preserved in our method.
> 3. KV cache eviction allows a simple way to trade efficiency and accuracy - increasing the kv size will guarantee that performance almost surely approaches original model performance. But with prompt compression, even with extremely long compressed prompt, no such guarantees can be made.
>
> -------------------------------------------------------------------------------------------------------------
> We thank the reviewer for their insightful comments. We hope that our response helps resolve the concerns, and we respectfully ask the reviewer to reconsider their assessment if they find our clarifications satisfactory.

---

> > ### Author Response · Authors · 2025-11-27
> > **Thank you for your feedback**
> >
> > Dear reviewer,
> > thank you again for your insightful and thoughtful comments. We have made our best efforts to give thorough explanations to your comments and updated the manuscripts accordingly.
> >
> > As we are now midway through the discussion period, we just wanted to politely follow up to see if our response has addressed your concerns. We are eager to engage further and would be very grateful for any additional feedback you may have. We sincerely hope you could look through our response and have a further comment at your convenience if you have any questions about the paper.
> >
> > Best wishes,
> > Authors.

---

### Official Review · Reviewer_TVsC · 2025-10-29

**Soundness:** 2
**Presentation:** 3
**Contribution:** 2
**Rating:** 4
**Confidence:** 3

**Summary:**

The paper introduces a learned KV cache management framework, LookaheadKV, that selectively evicts tokens during prefill stage by predicting future token importance using a small set of trainable tokens and LoRA adapters that are only active for these special tokens. The approach aims to achieve the accuracy benefits of similar draft-based approaches like LAQ and SpecKV, without their latency penalty from generating a draft response. LookaheadKV is trained with a normalized KL divergence loss and use the predicted importance to select top-$k$ KV cache entries. Experiments across various models show that LookaheadKV outperforms baselines and faster than draft-based approaches.

**Strengths:**

1. Presentation is overall clear and problem is motivated by the accuracy overhead trade-off from other draft-based methods. The latency problem is directly tackled through a learned approach. LookaheadKV provides an interesting application of parameter-efficient fine-tuning methodology for KV cache purposes.
2. Experiments performed on various model families and sizes, and also long-context benchmarks. Inclusion of LongProc provides some support that LookaheadKV can preserve the model's reasoning ability beyond recalling isolated facts.
3. Demonstrates efficiency by showing that method's impact on TTFT is almost negligible and competitive with SnapKV, while being significantly fast than both SpecKV and LAQ.

**Weaknesses:**

1. All ground-truth attention scores are generated using greedy decoding. As a result, lookaheadKV is effectively trained to predict attention patterns of a greedy future only, which may limit its applicability. In many practical applications, LLM inference is not deterministic and benefits immensely from stochastic sampling strategies such as temperature scaling to produce more diverse outputs. The attention patterns can differ substantially, which could lead to degradation in eviction quality when stochastic decoding is used such as in complex reasoning intensive tasks with longer thoughts.
2. Training lookaheadKV is expensive, since it requires for each model trained, to first generate a number of decoded tokens to use as ground truth.
3. The method defines ground-truth importance based on the attention of response tokens generated. However, attention scores do not always correspond to actual causal influence, so LookaheadKV may evict tokens whose attention is low but whose semantic contribution is high.
4. The need for requiring both lookahead tokens and lookahead lora is not made clear. While the ablation studies show the benefits, some additional justification is needed.

**Questions:**

1. Does inclusion of LoRA adapters enable using shorter length lookahead tokens? Could you achieve similar or better performance by simply increasing the number of lookahead tokens (e.g., n=64 or n=128) without LoRA adapters? What is the fundamental trade-off between lookahead token count and LoRA expressiveness in your framework?
2. How does lookaheadKV's performance vary when using stochastic decoding strategies at inference time?
3. The RULER evaluation (Figure 4, bottom) only reports results at budget=128, while LongBench shows multiple budgets (64, 256, 1024). How does LookaheadKV perform on RULER at these other budget settings, and are the performance advantages consistent across different budget constraints?
4. Results show from Table 3 that at n=32, performance mostly saturates. What happens to both performance and latency overhead when number of lookahead tokens exceeds 32 (e.g., 64, 128, 256)?

---

> ### Author Response · Authors · 2025-11-21
> **Official Comment by Authors (1/3)**
>
> Dear reviewer,
> We sincerely appreciate the time and effort you invested in reviewing our work.
>
> We are grateful that you recognized our proposed idea as an **interesting application** for KV cache purposes, acknowledged that our **experiments were performed across various models and benchmarks**, and highlighted our method’s **efficiency over prior methods**. Our responses to your comments are provided below.
>
> ### W1) All ground-truth attention scores are generated using greedy decoding. [...]
> We kindly refer the reviewer to our common response, where we provide a detailed discussion of LookaheadKV’s effectiveness during both training and inference under stochastic decoding settings. We thank the reviewer for raising this point.
>
> ### W2) Training lookaheadKV is expensive, since it requires for each model trained, to first generate a number of decoded tokens to use as ground truth.
>
> In general, our method requires for each model trained, to first generate responses to use as ground truth. However, as we show in Appendix section D, in cases where response generation is impractical, LookaheadKV can be trained using responses that are part of the original source data (e.g. ground-truth responses in SFT datasets) and skip response generation, with **almost no degradation in performance**. This suggests that, in scenarios where multiple models need to be trained, training them using the same set of responses from the original source data can be a cost-efficient and effective alternative.
>
> ### W3) The method defines ground-truth importance based on the attention of response tokens generated. [...]
> We acknowledge the reviewer's concern regarding attention scores potentially being a suboptimal measure of token importance. In our experiments, we adopt resposne token attention as the primary importance metric, because it is a well-established and standard metric validated by multiple prior works (SnapKV, SpecKV, LAQ, H2O, etc.) and our empirical results.
>
> We further highlight that our method is **compatible with alternative importance metrics**, as long as they can be computed. To demonstrate this flexibility, we trained LookaheadKV using the metric introduced in [1], where token importance is defined as the product of attention score and L1-norm of value vector: $s_{j} = \frac{1}{n_{\text{out}}} \sum_{i = n_{\text{in}} + 1}^{n_{\text{in}} + n_{\text{out}}} \mathbf{A}_{i,j} \times \|\mathbf{V}_j \|_1$. LongBench evaluations confirm comparable performance to using only attention as the importance metric.
>
> In summary, our framework can be easily adapted to various importance metrics, ensuring robustness of the approach.
>
> | Importance Metric          | narrativeqa | qasper | multifieldqa_en | hotpotqa | 2wikimqa | musique | gov_report | qmsum | multi_news | trec  | triviaqa | samsum | passage_count | passage_retrieval_en | lcc   | repobench-p | Avg. |
> |----------------------------|------------|--------|-----------------|----------|----------|---------|------------|-------|------------|-------|----------|--------|----------------|----------------------|-------|-------------|------|
> | Attention| 17.88| 12.93  | 39.31| 33.86    | 29.62    | 15.48   | 19| 20.16 | 19.2| 41.5  | 80.31| 34.39  | 2| 4.22| 31.51 | 32.56| 27.12 |
> | Attention + Value norm| 16.58| 13.28| 38.05| 33.66| 29.49| 15.73| 18.89| 20.36 | 19.15| 39.5  | 79.73| 34.99| 2.5| 3.95| 32.46 | 33.58| 26.99 |
>
> *LongBench Results using Various Importance Metrics on Llama3.2-1B-Instruct. Cache Budget=64*
>
> [1] [Attention Score is not All You Need for Token Importance Indicator in KV Cache Reduction: Value Also Matters.](https://aclanthology.org/2024.emnlp-main.1178/)

---

> ### Author Response · Authors · 2025-11-21
> **Official Comment by Authors (2/3)**
>
> ### W4) The need for requiring both lookahead tokens and lookahead lora is not made clear. [...]
> The main reason we use LoRA is to **improve the theoretical expressiveness** of the lookahead embeddings such that their queries can better predict the token importance scores. Only using lookahead tokens is limiting, because the original model weights are frozen and the entire attention patterns across all layers and heads must be learned by the small number of embedding parameters. While it might be theoretically possible to match the performance of using lookahead LoRA only by increasing the number of tokens, in practice, we would need a huge number of tokens to achieve the same performance, incurring intractable inference overhead.
>
> To demonstrate this, we extended the ablation results in Table 3 of our paper by evaluating at $n_{\text{lookahead}} = [64, 128]$. The results show that, after $n_{\text{lookahead}} = 32$, the performance saturates, while the inference overhead becomes significant due to increased number of tokens.
>
> In summary, lookahead LoRA provides 1) **orthogonal improvement** by allowing lookahead embeddings to learn richer intermediate representations, and 2) it further provides a **good balance of performance vs. overhead**, making our method practical.
>
> |            | **N=4 Score** | **N=4 Overhead (%)** | **N=8 Score** | **N=8 Overhead (%)** | **N=16 Score** | **N=16 Overhead (%)** | **N=32 Score** | **N=32 Overhead (%)** | **N=64 Score** | **N=64 Overhead (%)** | **N=128 Score** | **N=128 Overhead (%)** |
> |------------|------------|--------------------|------------|--------------------|--------------|----------------------|--------------|----------------------|--------------|----------------------|----------------|----------------------|
> | emb-only| 25.5| 3.4| 25.7| 3.8| 26.4| 3.4| 26.4| 4.2| 25.8| 7.3| 26.2| 10.7|
> | QV| 26.5| 3.7| 26.4| 4.1| 26.9| 4| 26.9| 4.4| 26.7| 7.7| 27.1| 10.9|
> | all| 26.6| 4.2| 27| 4.2| 27| 4.7| 27.1| 5| 27.1| 8.5| 27| 11|
>
> *Extended table 3 from the original paper. Results at n=[64, 128] are added.*
>
> We will make the need for lookahead LoRA clearer in the revised manuscript.
>
>
> ### Q1-1) Does inclusion of LoRA adapters enable using shorter length lookahead tokens?
> Yes. As we described in **W4**, using LoRA adapters in general enable using shorter length lookahead tokens while maintaining comparable performance.
>
> ### Q1-2) Could you achieve similar or better performance by simply increasing the number of lookahead tokens (e.g., n=64 or n=128) without LoRA adapters?
> Theoretically yes. With a very large N, one may be able to achieve similar performance. However, as described in **W4**, the performance improvement by increasing N satures at N=32, while the inference overhead increases significantly, making it impractical.
>
> ### Q1-3)What is the fundamental trade-off between lookahead token count and LoRA expressiveness in your framework?
> Identifying the precise trade-off of LoRA expressiveness and lookahead token count is indeed an interesting future work, but given the results in **W4**, we expect LoRA with lookahead tokens to always outperform using equal number of lookahead tokens only, and that a huge number of additional lookahead tokens would be needed to match the performance of using Lookahead LoRA.
>
> ### Q2) How does lookaheadKV's performance vary when using stochastic decoding strategies at inference time?
> As in **W1**, We kindly refer the reviewer to our common response, where we provide a detailed discussion of LookaheadKV’s effectiveness during both training and inference under stochastic decoding settings.

---

> > ### Author Response · Authors · 2025-11-21
> > **Official Comment by Authors (3/3)**
> >
> > ### Q3) How does LookaheadKV perform on RULER at other budget settings, and are the performance advantages consistent across different budget constraints?
> > Following your suggestion, we conduct RULER evaluation using Llama3.1-8B-Instruct across different budget configurations ranging from 64 to 2048.
> >
> > | Method / Budget    | 64    | 128   | 256   | 512   | 1024  | 2048  |
> > | :---        | :---  | :---  | :---  | :---  | :---  | :---  |
> > | FullKV      | 94.53 | 94.53 | 94.53 | 94.53 | 94.53 | 94.53 |
> > | SnapKV      | 27.80 | 48.96 | 67.89 | 74.49 | 85.24 | 94.07 |
> > | SpecKV      | $\underline{59.22}$ | $\underline{76.53}$ | 81.78 | 84.99 | 89.98 | 94.21 |
> > | LAQ         | 47.89 | 73.72 | **86.96** | **91.21** | **93.41** | **94.50** |
> > | LookaheadKV | **75.66** | **80.01** | $\underline{85.14}$ | $\underline{90.71}$ | $\underline{93.17}$ | $\underline{94.24}$ |
> >
> > *Table. RULER-4K average performance (%) over 13 tasks. (50 samples per task, total 650 samples)*
> >
> > As shown in the given table, across all budgets, **LookaheadKV is either comparable or superior**, especially at lower budgets, compared to all other baselines. As the budget increases, the performance gaps naturally narrow as they converge toward the FullKV upper bound, since most methods gain sufficient capacity to retain necessary context under relaxed constraints. On the other hand, we observe a significant performance advantage of LookaheadKV over other baselines under strict budget constraints. Our primary experiments in Figure 9 and Figure 10 further validate this strength, demonstrating robustness of LookaheadKV across diverse models and context lengths.
> >
> > ### Q4) What happens to both performance and latency overhead when number of lookahead tokens exceeds 32?
> > Yes, as shown in **W4**, the performance mostly saturates at n=32. When the number of lookahead tokens exceeds 32, the performance in general plateaus while the inference overhead increases significantly.
> >
> > -------------------------------------------------------------------------------------------------------------
> > We thank the reviewer for their insightful comments. We hope that our response helps resolve the concerns, and we respectfully ask the reviewer to reconsider their assessment if they find our clarifications satisfactory.

---

> > > ### Author Response · Authors · 2025-11-27
> > > **Thank you for your feedback**
> > >
> > > Dear reviewer,
> > > thank you again for your insightful and thoughtful comments. We have made our best efforts to give thorough explanations to your comments and updated the manuscripts accordingly.
> > >
> > > As we are now midway through the discussion period, we just wanted to politely follow up to see if our response has addressed your concerns. We are eager to engage further and would be very grateful for any additional feedback you may have. We sincerely hope you could look through our response and have a further comment at your convenience if you have any questions about the paper.
> > >
> > > Best wishes,
> > > Authors.

---

### Author Response · Authors · 2025-11-21
**Common Response Regarding Stochastic Decoding**

We sincerely thank the reviewers for their thoughtful and constructive feedback. Below, we provide a consolidated response to clarify the interplay between LookaheadKV and stochastic decoding, addressing concerns raised across multiple reviews.

## Interplay between LookaheadKV and Stochastic Decoding
We would first like to clarify that stochastic decoding can affect our method in **two distinct stages**:

**(1)** generating LookaheadKV training data using stochastic decoding *at train time*, and

**(2)** performing stochastic decoding *at inference time* after the LookaheadKV modules are trained and applied.

### Concerns regarding generating training data greedily
Regarding the concerns that LookaheadKV may be learning attention patterns specific to greedy future and may not generalize to stochastic future, we provide additional experimental results confirming that attention patterns under stochastic decoding **remain highly similar** to those under greedy decoding.

The table below shows the percentage of overlap of Top-512 tokens as well as the Kendall rank correlation coefficient between the importance scores induced by greedily decoding vs. stochastic decoding with varying temperature degrees. The scores are averaged over 30 randomly selected samples from our training data, across all layers and heads. Even when temerature is relatively high, (T= 0.8), we observe **strong persistence** of attention patterns. Notably, the deviation is **smaller** than that induced by using a speculative model (Llama3.2-1B-Instruct, equivalent to the SpecKV setting). This indicates that greedy-generated training data provides sufficiently robust learning signals for stochastic settings.

| Generation Method      | T = 0.2   | T = 0.4   | T = 0.6   | T = 0.8 | Llama3.2 1B, T = 0 |
|------------|-------|-------|-------|-------|----------|
| Top-512 Overlap (%)  | 95.06 | 93.73 | 91.40 | 91.37 | 88.66 |
| Kendall's Tau  | 91.44 | 88.63 | 84.61 | 84.79 | 80.05 |

*Importance Score Correlation measurement using Llama3.1-8B-Instruct. Various decoding methods are compared against greedy decoding.*

### Concerns regarding influence of stochastic decoding at inference
To further test LookaheadKV's performance under stochastic decoding during inference, we evaluate our method using two temperature settings: (0.2, 0.8). Results show that LookaheadKV maintains **superior performance over all other baselines** across all stochastic settings. Further, we observe that performance degradation at high temeprature setting (-4% at T=0.8) is consistent across all methods, including FullKV, confirming that stochasticity in inference affects all approaches similarly.

| Decoding Method | FullKV | LookaheadKV | SnapKV | LAQ | SpecKV |
|-------|--------|------|--------|-----|--------|
| **Greedy** | 49.88 | **47.72** | 43.50 | 46.61 | 45.45 |
| **T = 0.2** | 49.58 (-0.60 %) | **47.75 (+0.06 %)** | 43.29 (-0.48 %) | 46.73 (+0.26 %) | 44.99 (-1.01 %) |
| **T = 0.8** | 47.82 (-4.13 %) | **45.81 (-4.00 %)** | 41.39 (-4.85 %) | 45.27 (-2.87 %) | 43.43 (-4.44 %) |

*Average LongBench score using various decoding methods using Llama3.1-8B-Instruct. Cache Budget=128*


In summary, we show that 1) attention patterns **remain highly stable** under stochastic decoding, 2) performance degradation under high temperature is inherent to generation, not specific to LookaheadKV, and 3) LookaheadKV continues to **outperform all baselines** under non-greedy decoding settings.

We hope these additional analyses address the reviewers’ concerns and further clarify the robustness of LookaheadKV under stochastic decoding conditions. We sincerely appreciate the reviewers’ insightful comments, which helped strengthen the presentation of our work.

---

### Author Response · Authors · 2025-11-27
**To AC and Reviewers**

We sincerely thank the reviewers for the careful and thoughtful feedback.
We are encouraged that they commended our work as **an interesting application** (`TVsC`, `nWM2`) that **addresses a clearly defined and critical problem** (`TVsC`, `4uAb`), acknowledged our work as **a concise solution that is simple to integrate** (`nWM2`, `4uAb`), and highlighted **our comprehensive and rigorous experiments** (`TVsC`, `CrG9`, `nWM2`, `4uAb`) that **validates our method's strong empirical results in both performance** (`CrG9`, `nWM2`, `4uAb`) **and efficiency** (`TVsC`, `CrG9`, `nWM2`, `4uAb`). In response to the reviewer comments, we have made the following key additions to our paper. Major revisions are highlighted in blue in the new version.

# List of major updates to the paper
### 1. MT-Bench performance comparison across models and budgets (Table 2, Section 4.3)
### 2. Effect of stochastic decoding on LookaheadKV and baselines (Table 4, Section 5.2)
### 3. Extended ablation on trainable modules at N = [64, 128] (Table 5, Section 5.3)
### 4. Clarification on initialization of **1)** learnable lookahead tokens, **2)** the need for Lookahead LoRA, and  **3)** the TopK KV selection mechanism of LookaheadKV (Section 3.1)
### 5. Clarification on the compatibility of LookaheadKV training with other importance metrics (Section 3.2)
### 6. RULER evaluation on longer contexts (64K / 128K) (Table 6, Appendix E.1)
### 7. Effect of combining LookaheadKV and SnapKV-style suffix window (Table 7, Appendix E.2)
### 8. Discussion of generation stochasticity in LookaheadKV training (Table 8, Appendix E.3)
### 9. Complete per-task LongBench results (Table 9 - 14, Appendix E.4)

We again thank the reviewers for taking the time to share their thoughtful feedbacks. We are happy to answer any remaining questions they may have.

---

### Author Response · Authors · 2025-12-02
**Final Remark by Authors (1/2)**

We deeply regret the recent incident involving the identity leak and sincerely apologize to all reviewers and area chairs who have been adversely affected. We especially acknowledge the additional burden and responsibilities this situation has placed on the ACs, and we truly appreciate their dedication under these challenging circumstances.

We also find it deeply unfortunate that, although we believe we have effectively addressed the reviewers' concerns, we were unable to receive any responses from reviewers due to the unexpectedly shortened rebuttal period. We kindly request that this context be considered when evaluating the merits of our paper.

Once again, we express our sincere apologies for this unfortunate incident and our gratitude for the time and effort invested by the ACs.

--------

We again sincerely thank the reviewers for the careful and thoughtful feedback. We are encouraged that they commended our work as an **interesting and novel application** (TVsC, nWM2) for KV cache management that **addresses a clearly defined and critical problem** (TVsC, 4uAb), acknowledged our work as a **concise solution that is simple to integrate** (nWM2, 4uAb), and **highlighted our comprehensive and rigorous experiments** (TVsC, CrG9, nWM2, 4uAb) that validates our method's **strong empirical results in both performance** (CrG9, nWM2, 4uAb) and **efficiency** (TVsC, CrG9, nWM2, 4uAb).

Below, we summarize the key concerns raised and how we address them:

> 1. Unclear effect of stochastic decoding on LookaheadKV training & inference (TVsC, CrG9, 4uAb)

We provided:

- additional LongBench experiments showing that our method does not harm stochastic decoding performance after eviction, and obtains **the best performance compared to all baselines** under all tested stochastic generation settings. (Section 5.2, Table 4)
- additional experiments on importance score similarity of greedy response against stochastic responses. Results show that importance scores of input tokens remain **highly stable**, regardless of whether response was generated greedily vs. stochastically. (Section E.3, Table 8), indicating that token importance is **largely unaffected** by the method used to generated responses.

> 2. Limitation of using attention score of response as importance metric (TVsC)

We clarified that:

1. attention score is an effective metric for measuring token importance that is validated in most popular prior KV cahce eviction papers (SnapKV, PyramidKV, H2O, SpecKV, LAQ, etc.) as well as our own experiments
2. more importantly, our method is **compatible with any other token-wise importance metric**. **We prove this through additional experiments**, where we do LookaheadKV training using importance metric defined in [1], showing the resulting model's performance is comparable with our method. (Section 3.2, Rebuttal of reviewer TVsC, **W3**)

[1] [Attention Score is not All You Need for Token Importance Indicator in KV Cache Reduction: Value Also Matters.](https://aclanthology.org/2024.emnlp-main.1178/)

> 3. Evaluation under multi-turn conversation setting (CrG9)

We provided **the complete MT-Bench evaluation results across all models with varying cache budgets.** Again, our method **performs on par or outperforms all baselines** across all models and all budgets. (Section 4.3, Table 2)

> 4. Unclear presentation regarding initialization of lookahead token & the need for lookahead LoRA (TVsC, nWM2)

We clarified:

1. the need for Lookahead LoRA rather than only using the lookahead tokens (Section 3.1, Rebuttal of reviewer TVsC, **W4**). Specifically, **we provided additional ablation of Table 5** to show that by only using the lookahead tokens, the performance quickly saturates to a suboptimal point while incurring significant inference overhead.
2. how the lookahead token embeddings are obtained. We explained that the lookahead tokens are **new tokens added to the vocabulary, and that their embeddings are randomly initialized before training.** (Section 3.1, Rebuttal of reviewer nWM2, **W1**)

> 5. Missing per-task LongBench scores (CrG9)

We provided **the complete per-task LongBench evaluation results across all models and all budgets**. (Section E.4, Table 9-14)

> 6. Analysis on combining SnapKV's observation window with LookaheadKV (nWM2)

We provided **additional experimental results combining SnapKV's observation window** (Section E.2, Table 7). The slight drop in the overall performance when combining SnapKV's window indicates that **LookaheadKV's observation window is superior for more accurate importance estimation**.

---

> ### Author Response · Authors · 2025-12-02
> **Final Remark by Authors (2/2)**
>
> > 7. Analysis on longer context length (nWM2)
>
> We provided additional experimental results by evaluating on RULER at 64K and 128K (Section E.1, Table 6). Our method **continues to outperform all baselines at these contexts as well**, surpassing competitive draft-based approaches by up to **8.3%**
>
> > 8. Justification of training cost overhead vs. inference efficiency (TVsC, nWM2, 4uAb)
>
> We emphasized that:
>
> 1. **LookaheadKV is resource-saving.** As acknowledged by reviewer `4uAb`, LookaheadKV training cost can be amortized since it is a **one-time effort that significantly reduces the inference-time overhead**, similar to how one would perform SFT to adapt a model to a specific task rather than using e.g., a few-shot prompt for each inference call, to reduce the prefill cost.
> 2. **it substantially reduces TTFT (by nearly x2 faster)**, delivering a clearly improved user experience. (Section 5.1, Table 3)
> 3. **LookaheadKV outperforms baselines.** Our method not only reduces resource usage, but it also performs consistently better than all baselines.  (Section 4.3, Figure 4)
>
> ---------
>
> We would like to once again extend our gratitude to the reviewers for their thoughtful feedback. Also, we sincerely appreciate the time and effort that the Area Chair has invested in evaluating our submission, especially under the extraordinary circumstances surrounding the recent incident.
>
> Best regards, Authors.

---

### Meta-Review · Area_Chair_EhXs · 2026-01-11

**Summary:**

This paper proposes LookaheadKV, a framework for KV cache eviction in LLMs. The core innovation lies in using a set of learnable 'lookahead tokens' and specialized 'Lookahead LoRA' modules to predict future attention patterns during the prefill stage. This approach effectively circumvents the high latency costs associated with draft-based methods while maintaining superior accuracy compared to heuristic methods. While the performance gains compared to training-free methods are relatively modest, the overall idea is novel and presents an interesting new solution for KV cache eviction. Despite the initially negative ratings, the authors provided a detailed and effective rebuttal that resolved most of the reviewers' concerns. Therefore, I recommend accepting this paper.

**Reviewer Concerns:**

Stochastic decoding: The authors provided new experiments demonstrating robust performance.

Training cost: The one-time training cost is acceptable considering the significant improvements achieved in inference efficiency.

Evaluation scope: The authors expanded the evaluation scope to include multi-turn dialogue and ultra-long context scenarios.

Remaining limitations: A limitation persists regarding the method's current focus exclusively on prefill-stage eviction.

**Reviewer Scores:**

4,4,4,8 ->6,6,6,8

---

### Decision · Program_Chairs · 2026-01-26

Accept (Poster)